# Carbon-nitrogen interactions in idealized simulations with JSBACH (version 3.10)

Daniel S. Goll[1,2], Alexander J. Winkler[3,4], Thomas Raddatz[3], Ning Dong[5,6], Ian Colin Prentice[5,7], Philippe Ciais[1], and Victor Brovkin[3]

[1]Le Laboratoire des Sciences du Climat et de l'Environnement, IPSL-LSCE CEA/CNRS/UVSQ Saclay, Gif sur Yvette, France.
[2]also guest scientist at Max Planck Institute for Meteorology, Hamburg, Germany
[3]Max Planck Institute for Meteorology, Hamburg, Germany
[4]International Max Planck Research School on Earth System Modelling, Hamburg, Germany
[5]Department of Biological Sciences, Macquarie University, North Ryde, NSW 2109, Australia
[6]Faculty of Agriculture and Environment, Department of Environmental Sciences, University of Sydney, NSW 2006, Australia
[7]AXA Chair in Biosphere and Climate Impacts, Department of Life Sciences, Imperial College London, Silwood Park Campus, Buckhurst Road, Ascot SL5 7PY, UK

*Correspondence to:* Daniel S. Goll (daniel.goll@lsce.ipsl.fr)

**Abstract.** Recent advances in the representation of soil carbon decomposition and carbon-nitrogen interactions implemented previously into separate versions of the land surface scheme JSBACH are here combined in a single version which is set to be used in the upcoming $6^{th}$ phase of coupled model intercomparison project (CMIP6).

Here we demonstrate that the new version of JSBACH is able to reproduce the spatial variability in the reactive nitrogen loss pathways as derived from a compilation of $\delta^{15}$N data (R=.76, RMSE=.2, Taylor score=.83). The inclusion of carbon-nitrogen interactions leads to a moderate reduction (-10%) of the carbon-concentration feedback ($\beta_L$) and has a negligible effect on the sensitivity of the land carbon cycle to warming ($\gamma_L$) compared to the same version of the model without carbon-nitrogen interactions in idealized simulations (1% increase in atmospheric carbon dioxide per yr). In line with evidence from elevated carbon dioxide manipulation experiments, pronounced nitrogen scarcity is alleviated by (1) the accumulation of nitrogen due to enhanced nitrogen inputs by biological nitrogen fixation and reduced losses by leaching and volatilization. Warming stimulated turnover of organic nitrogen further counteracts scarcity.

The strengths of the land carbon feedbacks of the recent version of JSBACH, with $\beta_L = 0.61$ Pgppm$^{-1}$ and $\gamma_L = -27.5$ Pg$^{\circ}$C$^{-1}$, are 34% and 53% less than the averages of CMIP5 models, although the CMIP5 version of JSBACH simulated $\beta_L$ and $\gamma_L$ which are 59% and 42% higher than multi-model average. These changes are primarily due to the new decomposition model, indicating the importance of soil organic matter decomposition for land carbon feedbacks.

## 1 Introduction

The version of the Max-Planck-Institute Earth System Model (MPI-ESM) used in the $5^{th}$ phase of the coupled model intercomparison project (CMIP5) experienced pronounced biases in simulated soil carbon (Todd-Brown et al., 2013), soil hydrology

(Hagemann and Stacke, 2014), and the lack of carbon-soil nutrient interactions (Zaehle et al., 2014; Wieder et al., 2015), hampering the reliability of the simulated response of land system to increasing carbon dioxide ($CO_2$), climate and land use and land cover changes. Recent model developments addressed these issues (Goll et al., 2012, 2015; Hagemann and Stacke, 2014) in separate versions of the land surface scheme of the MPI-ESM, JSBACH, but have not been yet combined in a single model version.

The projected carbon balance in JSBACH was substantially affected by recent model developments: The implementation of carbon-, nitrogen- and phosphorus interactions reduced accumulated land carbon uptake by 25% between 1860–2100 under a business as usual scenario (Goll et al., 2012), while the implementation of a new decomposition model (YASSO) reduced the accumulated land carbon uptake by about 60% in the same period (Goll et al., 2015). The exchange of the former CENTURY
type soil decomposition model (Parton et al., 1993) with the YASSO decomposition model (Tuomi et al., 2008, 2009, 2011) improved the present-day state of the carbon cycle compared to observations as well as the response of decomposition to soil warming, and substantially reduced the uncertainties of land use change emissions for a given land use change scenario (Goll et al., 2015). The strong impact on the carbon balance of each of both developments underline the importance of combining them in a single version.

The capacity of land ecosystems to increase their nitrogen storage as well as to enhance recycling of nitrogen in organic matter are major constraints on their ability to increase carbon storage under elevated $CO_2$ concentrations (Hungate et al., 2003; Thomas et al., 2015; Liang et al., 2016). The respective response patterns of nitrogen processes governing the balance and turnover of organic nitrogen are crucial (Niu et al., 2016) to asses the likelihodd of the occurence of (progressive) nitrogen limitation (Luo et al., 2004). Recent advances in the interpretation of soil $\delta^{15}N$ global data sets provide a promising tool
(Houlton et al., 2015; Zhu and Riley, 2015) by allowing a more detailed evaluation of the nitrogen loss pathways in land carbon-nitrogen models than previously done (e.g. Parida (2011); Goll et al. (2012)).

    Since future scenarios of $CO_2$ concentrations differ among CMIP phases, an idealized setup of an annual increase in $CO_2$ concentration by 1% is used to foster the analysis of the carbon cycle feedbacks among models, and to compare emerging properties of different model versions in various CMIPs (Eyring et al., 2016). We adopt this approach taking advantage of
25 existing simulations of climatic changes in this idealized setup of the CMIP5 intercomparison (Arora et al., 2013) to drive the land surface model JSBACH uncoupled from the atmosphere and ocean components of the earth system model.

    This article documents the modifications to the soil carbon decomposition (Goll et al., 2015) and nitrogen cycle (Parida, 2011; Goll et al., 2012) submodels and the combination of both developments in a recent version of JSBACH including an advanced soil hydrological scheme (Hagemann and Stacke, 2014), scheduled to be used in CMIP6. We further analyzed the
30 state of the nitrogen cycle using soil $\delta^{15}N$ data and quantified the carbon cycle feedbacks to increasing $CO_2$ concentrations and climate change. The analysis aims at facilitating the interpretation of the models dynamics in the upcoming round of CMIP experiments (Eyring et al., 2016), and allows a straightforward comparison to the result from the previous round of CMIP (Taylor et al., 2012).

## 2 Methods

### 2.1 Model description

The implementation of the nitrogen cycle and the soil carbon and litter decomposition model YASSO is described in detail in Parida (2011); Goll et al. (2012) and Goll et al. (2015), respectively. In the following, a brief summary of the major concepts is given and afterwards the modifications to the original developments needed to combine them are documented in detail. The notation applied here follows Goll et al. (2012, 2015) and a scheme of the cycling of carbon and nitrogen as well as their interactions are given in Figure 1.

The decomposition model (YASSO) is based on a compilation of litter decomposition and soil carbon data and distinguish organic matter fractions according to litter size and solubility (Tuomi et al., 2008, 2009, 2011). In JSBACH we use two litter size classes, which correspond to litter from non-lignified and lignified plant material (Goll et al., 2015). Each of the two litter classes is further refined into four solubility classes (acid-soluble ($C_A$), water-soluble ($C_W$), ethanol-soluble ($C_E$), nonsoluble ($C_N$)) (Eq. 1). One additional pool ($C_H$) represents humic, slowly-decomposing substances.

The interactions between nitrogen availability and carbon fluxes, namely primary productivity and decomposition, are based on the concept of $CO_2$-induced nutrient limitation (CNL) (Goll et al., 2012). In this framework, we distinguish between CNL and background nutrient limitation. The latter is assumed to be indirectly considered in the original parametrization of carbon cycle processes as they are based on measurements in present day ecosystems and therefore reflect present day nutrient conditions. CNL is an additional nutrient limitation caused by the increase in atmospheric $CO_2$ and is computed dynamically according to nutrient supply and demand. In case microbial and vegetation nitrogen demand cannot be met by the supply, all carbon fluxes of which the donor compartment has a higher C:N ratio than the receiving pool (i. e. the fluxes of carbon from the solubility classes pools to the humus pool) are down-regulated. The concept of CNL allows to introduce carbon-nitrogen interactions to YASSO, as the needed conditions are met, e.g. the parametrization of YASSO indirectly reflects present day nutrient effects on decomposition as it is based on leaf litter experiments.

Following Goll et al. (2012), CNL affects the decomposition of all pools except the slowly-decomposing nutrient-rich pool (Eqs. 2–5). The litter decomposition data on which YASSO is based is not suited to link the fate of nitrogen in litter to the respective solubility pools. Therefore, we assume one single nitrogen pool representing all nitrogen linked to the four carbon solubility pools per litter class (Eq. 7). This can be refined in the future if appropriate data becomes available.

The nitrogen cycling is primarily driven by carbon fluxes using constant N-to-C ratios of organic pools (Eqs. 6–7), with the exceptions of the non-lignified litter pool (Eq. 8) (Parida, 2011). Further exceptions are the processes linking the terrestrial carbon cycle with the atmosphere (biological nitrogen fixation and denitrification) and the aquatic systems (leaching) which are computed either as substrate-limited (Eqs. 16–17) or, for the case of biological nitrogen fixation (Eq. 15), as driven by demand due to the ample supply of $N_2$ in the atmosphere (Parida, 2011). The nitrogen cycle and its interactions with the carbon cycle are not modified. The only exception is that the turnover times of the nitrogen litter and soil organic matter pools are derived from the YASSO decomposition model (Eq. 10) instead of the former decomposition model.

All parameters and variables are given in Table 1&2

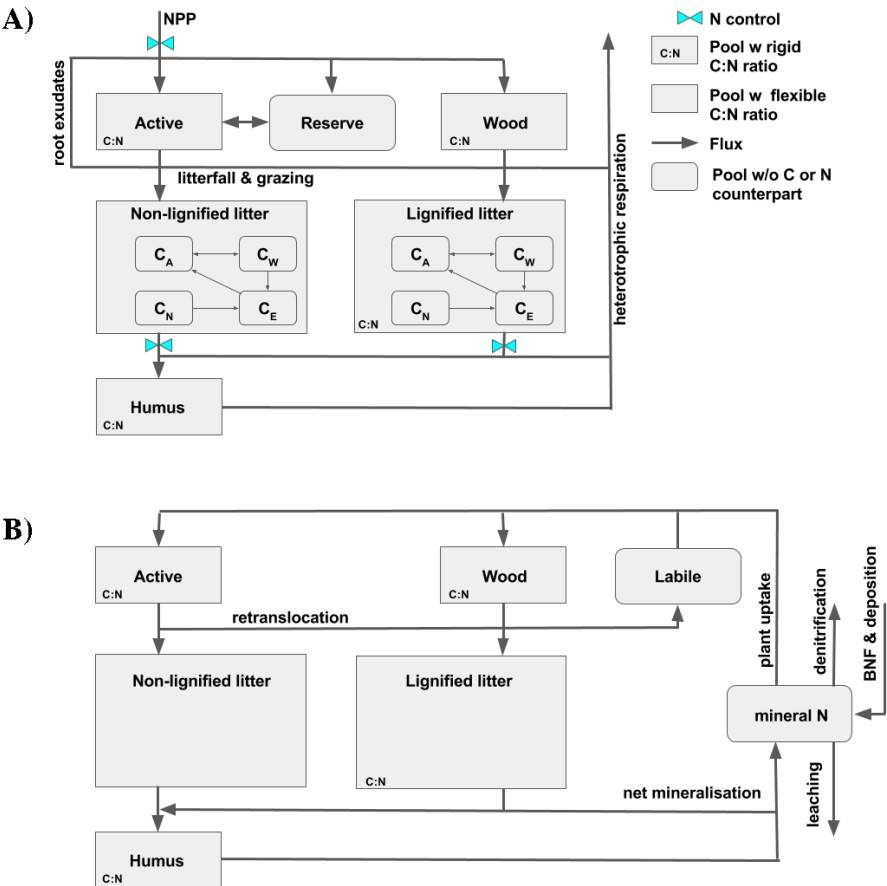

**Figure 1.** Schematic representation of carbon (A) and nitrogen (B) cycling in JSBACH. Vegetation is represented by four pools: "active" (leaves and non-lignified tissue) and "wood" (stem and branches), "reserve" (sugar and starches) and "labile" (mobile nitrogen) (Goll et al., 2012). Dead organic matter is represented by "non-lignified litter", "lignified litter" (lignified litter and fast-decomposing soil organic matter), and "humus" (slow-decomposing organic matter) (Raddatz et al., 2007). All organic matter pools have fixed C:N ratios, except the pools "reserve", "labile" and "non-lignified litter". While the first two pools have no corresponding pool, the C:N ratio of the latter pool varies according the balance between immobilization demand and supply. The carbon in the litter compartment is further refined into the acid-soluble (A), water-soluble (W), ethanol-soluble (E), and non-soluble (N) compounds (Goll et al., 2015) which have no C:N ratio assigned. Soil mineral nitrogen is represented by a single pool (soil mineral pool). The opposing triangle marks carbon fluxes which are downregulated in case the nitrogen demand exceeds the nitrogen supply.

### 2.1.1 Nitrogen effect on decomposition

Matrix $\mathbf{C}$ describes the soil carbon pools $(A, W, E, N)$ of the two litter classes $(i)$ in JSBACH, excluding recalcitrant humic substances $(C_H)$:

$$\mathbf{C_i} = \begin{pmatrix} C_{A,i} \\ C_{W,i} \\ C_{E,i} \\ C_{N,i} \end{pmatrix} \tag{1}$$

5     The dynamics of the soil carbon pools are described as

$$\frac{d\mathbf{C_i}}{dt} = \mathbf{A_p k_i(F) C_i} + \mathbf{I_i} \tag{2}$$

where $\mathbf{A_p}$ is the mass flow matrix; $\mathbf{k_i(F)}$ is a diagonal matrix of the decomposition rates $\mathbf{k_i(F)} = diag(k_{A,i}, k_{W,i}, k_{E,i}, k_{N,i})(\mathbf{F})$ as a function of climatic conditions $(\mathbf{F})$; and matrix $\mathbf{I_i}$ is the carbon input of type $i$ to the soil. The dynamics of the humus pool $(C_H)$ are described as

$$\frac{dC_H}{dt} = p_H \sum_{i=1}^{N} \mathbf{k_i(F) C_i} - k_H C_H \tag{3}$$

where $p_H$ is the relative mass flow parameter and $k_H$ the decomposition rate of the humus pool. A detail description of decomposition can be found in the supplementary information of Goll et al. (2015), here we only focus on the modification of the original implementation.

    The climate dependence of the decomposition rate factor $k_{j,i}$ of the carbon pools was originally implemented by Goll et al.
(2015) based on Tuomi et al. (2008) as:

$$k_{j,i}(\mathbf{F}) = \alpha_{j,i} exp(\beta_1 T + \beta_2 T^2)(1 - exp(\gamma P)), \tag{4}$$

where $T$ is air temperature and $P$ is precipitation, $\beta_1$, $\beta_2$, $\gamma$ are parameters, and $\alpha_{j,i}$ are decomposition rates at references conditions $(T = 0$ and $P \to \infty)$ of pool $i$ of litter class $j$. $\alpha_{j,i}$ is the product of reference decomposition rate $r_j$ of solubility class $j$ and the litter diameter $d_i$ of litter class $i$. YASSO uses precipitation instead of the more direct driver of soil moisture
due the lack of adequate soil moisture observation to relate the decomposition date the model is based on (Tuomi et al., 2008).

    We introduced a scaling factor, namely the nitrogen limitation factor $(f_{limit}^N)$, to account for the down-regulation of decomposition when nitrogen is in short supply

$$k_{j,i}(\mathbf{F}) = f_{limit}^N \alpha_{j,i} exp(\beta_1 T + \beta_2 T^2)(1 - exp(\gamma P)), \tag{5}$$

$f_{limit}^N$ is calculated based on a supply and demand approach (Parida, 2011; Goll et al., 2012). In a first step, potential carbon
fluxes are computed from which the gross mineralisation, immobilization and plant uptake of mineral nitrogen is diagnosed. In a second step, all fluxes consuming nitrogen (donor compartment has a higher C:N ratio than the receiving pool as well as plant

**Table 1.** Variables of the model.

| Variable | Units | Description |
|---|---|---|
| $P$ | $\mathrm{mday}^{-1}$ | 30 days average of daily precipitation |
| $T$ | $^\circ\mathrm{C}$ | 30 days average of daily 2m air temperature |
| $C_{j,i}$ | $\mathrm{mol(C)m}^{-2}$ | soil organic carbon of solubility class $j$ and litter class $i$ |
| $N_x$ | $\mathrm{mol(N)m}^{-2}$ | nitrogen in compartment $x$ |
| $\overline{NPP}$ | $\mathrm{mol(C)m}^{-2}\mathrm{day}^{-1}$ | annual average of daily net primary productivity |
| $F^{C}_{x \triangleright y}$ | $\mathrm{mol(C)m}^{-2}\mathrm{day}^{-1}$ | daily flux of carbon from compartment $x$ to compartment $y$ |
| $F_{extr}$ | $\mathrm{mol(C)m}^{-2}\mathrm{day}^{-1}$ | daily flux of nitrogen from/to the atmosphere and to aquatic systems |
| $F_{leach}$ | $\mathrm{mol(C)m}^{-2}\mathrm{day}^{-1}$ | daily flux of nitrogen leached to aquatic systems |
| $F_{denit}$ | $\mathrm{mol(C)m}^{-2}\mathrm{day}^{-1}$ | daily flux of nitrogen lost to the atmosphere by denitrifcation |
| $BNF$ | $\mathrm{mol(N)m}^{-2}\mathrm{day}^{-1}$ | daily nitrogen inputs by biological nitrogen fixation |
| $D_x$ | $\mathrm{mol(N)m}^{-2}\mathrm{day}^{-1}$ | nitrogen demand of vegetation ($x = veg$) or immobilization ($x = micr$) |
| $f^{N}_{limit}$ | – | nitrogen limitation factor |
| $d_x$ | $\mathrm{day}^{-1}$ | decomposition rate of nitrogen in compartment $x$ |
| $k_{j,i}$ | $\mathrm{day}^{-1}$ | decomposition rate of solubility class $j$ and litter class $i$ |
| $k_H$ | $\mathrm{day}^{-1}$ | decomposition rate of humus pool |
| $\epsilon$ | $\mathrm{day}^{-1}$ | consumption rate by herbivores |
| $f_{h2o}$ | – | daily fraction of soil water lost due to runoff and drainage |
| $\alpha$ | – | soil moisture stress on biological processes |

uptake) are down-regulated in case nitrogen demand cannot be met by the nitrogen supply. Hereby, a common scalar ($f^{N}_{limit}$) (see appendix) is used thereby no assumption about the relative competitive strengths of microbial and plant consumption has to be made. In case nitrogen demand is met by the supply, the fluxes computed in the first step are taken as actual ones without any modification.

## 2.1.2 Dynamics of nitrogen in litter and soil organic matter

Nitrogen in litter and soil organic matter is separated into three pools, namely slowly-decomposing organic matter (humus) $C_H$, lignified litter and fast decomposing organic matter $C_{l,w}$, as well as non-lignified litter and fast decomposing organic matter $C_{l,a}$ (Goll et al., 2012). We assigned each of the three nitrogen pools to one or more corresponding YASSO pools (Table 3). A refinement of the representation of nitrogen in decomposing material following strictly the carbon classification is not straightforward as the carbon pools ($A, W, E, N$) defined by their respective solubility characteristics do not correspond to substance classes with distinguished stoichiometries.

**Table 2.** Parameters of the model. [*] $\alpha_{j,i}$ is an array and values can be found in Tuomi et al. (2011).

| Parameter | Value | Units | Description | Source |
|-----------|-------|-------|-------------|--------|
| $\beta_1$ | 0.095 | $°\text{C}^{-1}$ | Temperature dependence of decomposition | Goll et al. (2015) |
| $\beta_2$ | $-1.4 \times 10^{-3}$ | $°\text{C}^{-2}$ | Temperature dependence of decomposition | Goll et al. (2015) |
| $\gamma_1$ | -1.21 | $\text{m}^{-1}$ | Precipitation dependence of decomposition | Goll et al. (2015) |
| $\alpha_{j,i}$ | [*] | $\text{day}^{-1}$ | decomposition rates at references conditions | |
| | | | ($T = 0$ and $P \to \infty$) of pool $i$ of litter class $j$ | Tuomi et al. (2011) |
| $\alpha_H$ | $4.383 \times 10^{-6}$ | $\text{day}^{-1}$ | decomposition rates at references conditions | |
| | | | ($T = 0K$ and $P \to \infty$) of humus pool | Tuomi et al. (2011) |
| $r_s$ | 10. | $\text{mol(N)mol}^{-1}\text{(C)}$ | N-to-C ratio of slowly decomposing organic matter | Goll et al. (2012) |
| $r_{lw}$ | 330. | $\text{mol(N)mol}^{-1}\text{(C)}$ | N-to-C ratio of lignified litter | Goll et al. (2012) |
| $r_{la}$ | 55. | $\text{mol(N)mol}^{-1}\text{(C)}$ | N-to-C ratio of non-lignified litter | Goll et al. (2012) |
| $r_w$ | 150. | $\text{mol(N)mol}^{-1}\text{(C)}$ | N-to-C ratio of lignified biomass | Goll et al. (2012) |
| $\beta_3$ | 0.7 | – | fraction of nitrogen in excrement in labile form | Parida (2011) |
| $f_{emp}$ | $-3.0 \times 10^{-3}$ | $\text{dayg}^{-1}\text{(C)}$ | NPP dependence of biological nitrogen fixation | Cleveland et al. (1999) |
| $f_{bnf}$ | 0.7 | $\text{g(N)m}^{-2}\text{day}^{-1}$ | scaling factor of biological nitrogen fixation | this study |
| $f_s$ | 0.1 | – | fraction of soil mineral nitrogen in soil solution | this study |
| $k_{denit}$ | $2.0 \times 10^{-3}$ | – | daily fraction of soil mineral nitrogen lost by denitrifcation | Parida (2011) |
| $w_C$ | 12.011 | $\text{g(N)m}^{-2}\text{day}^{-1}$ | standard atomic weight of carbon | |
| $w_N$ | 14.007 | $\text{g(N)mol}^{-1}\text{(N)}$ | standard atomic weight of nitrogen | |
| $t$ | 1 | day | time step | |

**Table 3.** The nitrogen pools and the corresponding carbon pools for humus ($H$) and lignified (woody) ($w$) and non-lignified (active) ($a$) plant material.

| Nitrogen | Carbon |
|----------|--------|
| $N_H$ | $C_H$ |
| $N_{l,w}$ | $C_{A,w} + C_{W,w} + C_{E,w} + C_{N,w}$ |
| $N_{l,a}$ | $C_{A,a} + C_{W,a} + C_{E,a} + C_{N,a}$ |

In JSBACH, nitrogen in compartments with a fixed N-to-C ratio, namely nitrogen in lignified litter ($N_{l,w}$) as well as nitrogen in slowly decomposing organic matter ($N_H$), are derived from the corresponding YASSO carbon pools ($C_{j,i}$) by:

$$N_H = r_H C_H \tag{6}$$

$$N_{l,w} = r_{lw}(C_{A,w} + C_{W,w} + C_{E,w} + C_{N,w}) \tag{7}$$

where $r_H$ is the N-to-C ratio of former slow carbon pool ($C_H$) now applied to the humus pool of YASSO, and $r_{lw}$ of former lignified (woody) litter pool ($C_{l,w}$) now applied to the sum of the solubility class pools for lignified litter of YASSO.

The dynamics of nitrogen in non-lignified litter & fast decomposing organic matter ($N_{l,a}$) were not modified from the original nitrogen-enabled version of JSBACH (Parida, 2011) and are given by:

$$\frac{dN_{l,a}}{dt} = r_{la}F^C_{a \triangleright la} + (1 - \beta_3)\epsilon N_a - f^N_{limit}d_{la} * N_{l,a} \tag{8}$$

where the first term describes the nitrogen influx from active, non-lignified plant tissue ($N_a$), the second term describes the flux of nitrogen from herbivores excrements which is not directly available to biota, and the third term arises from the nitrogen released by biological mineralization of litter and fast-decomposing soil organic matter. We assume that active plant material ($N_a$) is consumed by herbivores at a constant rate ($\epsilon$) and immediately excreted (Parida, 2011). We separate the excrement into fast decomposing ($1 - \beta_3$) and labile ($\beta_3$) nitrogen compounds, the first enters the non-lignified litter pool ($N_{l,a}$) and the latter the soil mineral nitrogen pool (Equation 11).

The decomposition rate $d_{la}$ of nitrogen in litter and fast decomposing soil organic matter equals the decomposition rate of the sum of the YASSO carbon pools $C_{A,a} + C_{W,a} + C_{E,a} + C_{N,a}$ and is given by

$$\frac{d(C_{A,a} + C_{W,a} + C_{E,a} + C_{N,a})}{dt} = d_{la}(C_{A,a} + C_{W,a} + C_{E,a} + C_{N,a}) \tag{9}$$

so that $d_{la}$ can be derived from

$$d_{la} = \frac{C_{A,a}(t+1) + C_{W,a}(t+1) + C_{E,a}(t+1) + C_{N,a}(t+1)}{C_{A,a}(t) + C_{W,a}(t) + C_{E,a}(t) + C_{N,a}(t)} - 1 \tag{10}$$

As we calculate potential decomposition fluxes in a first step to derive nitrogen demand (see Goll et al. 2012) we know the state of pools for time $t$ and $t+1$.

The dynamics of the soil mineral nitrogen ($N_{smin}$) were not modified and are given - as originally formulated by Parida (2011) - by:

$$\frac{dN_{smin}}{dt} = F_{extr} + \beta_3\epsilon N_a + d_H N_H + (r_w - r_{lw})F^C_{w \triangleright lw} - f^N_{limit}(D_{veg} + D_{micr}) \tag{11}$$

where $F_{extr}$ is the net of fluxes connecting the compartments considered in the model and outside (here: biological dinitrogen ($N_2$) fixation, leaching, $N_2$ and nitrous oxide ($N_2O$) emissions (Equation 14)), $\beta_3\epsilon N_a$ is flux of labile nitrogen from herbivores

excrements, and $D_{micr}$ and $D_{veg}$ are the nitrogen demands of vegetation and microbes, respectively. Due to the lower nitrogen content of litter compared to humus, the decomposition of lignified and non-lignified litter corresponds to a net immobilization of nitrogen, which is part of the $D_{micr}$. The term $(r_w - r_{lw})F^C_{w \triangleright lw}$ represents nitrogen leaching from freshly shedded wood given by the decomposition and the stoichiometries assigned to wood ($r_w$) and lignified litter ($r_{lw}$). The decomposition rate of nitrogen in the humus pool ($N_H$), $d_H$, equals the decomposition rate of the corresponding YASSO carbon pool ($C_H$), $k_H$. This rate according to Eq.(4) is given by:

$$k_H(\mathbf{F}) = \alpha_H exp(\beta_1 T + \beta_2 T^2)(1 - exp(\gamma P)), \tag{12}$$

Note that there is no nutrient effect on the decomposition of $N_H$ and $k_H$ is calculated exactly like described in Goll et al. (2015).

For the calculation of the microbial (soil) nutrient demand ($D_{micr}$) we substituted the pools $C_{l,a}$ and $C_H$ with the corresponding YASSO pools in Eq.(15) of Goll et al 2012:

$$D_{micr} = (r_H - \frac{N_{l,a}}{(C_{A,a} + C_{W,a} + C_{E,a} + C_{N,a})})F^C_{la \triangleright s} + (r_H - r_{lw})F^C_{lw \triangleright s} - \frac{N_{l,a}}{(C_{A,a} + C_{W,a} + C_{E,a} + C_{N,a})})F^C_{la \triangleright a} - r_{lw}F^C_{lw \triangleright a} \tag{13}$$

The fluxes $F^C_{la \triangleright s}$ and $F^C_{lw \triangleright s}$ are the net fluxes of carbon to the humus from the solubility pools (AWEN) of non-lignified and lignified litter, respectively. $F^C_{la \triangleright a}$ and $F^C_{lw \triangleright a}$ are the respective sums of respiration fluxes of the AWEN pools.

## 2.2 The processes governing the terrestrial nitrogen balance in JSBACH

Nitrogen enters terrestrial ecosystems by biological nitrogen fixation ($BNF$), as well as atmospheric deposition, while nitrogen is lost via leaching ($F_{leach}$), erosion (omitted in JSBACH) and denitrification ($F_{denit}$):

$$F_{extr} = BNF - F_{leach} - F_{denit} \tag{14}$$

BNF in global models is commonly represented by an empirical relationship based on a compilation of site measurements (Cleveland et al., 1999). Due to the lack of a process-based alternatives, we use this approach as described in Parida (2011) despite its shortcomings (Thomas et al., 2013; Sullivan et al., 2014; Wieder et al., 2015). In this approach BNF ($BNF$) is derived from the annual average of daily net primary productivity ($\overline{NPP}$) using the empirical relationship between BNF and evapotranspiration (Thornton et al., 2007):

$$BNF = (f_{bnf} * (1 - e^{(f_{emp} * w_C \overline{NPP})})\frac{w_N}{w_C} \tag{15}$$

where $f_{emp} = -0.003 \text{ dayg}^{-1}(C)$ is an empirical relationship from Cleveland et al. (1999) , $f_{bnf} = 0.7 \text{ g(N)m}^{-2}\text{day}^{-1}$ is a calibrated constant to achieve a global sum of BNF of 100 $\text{Mtyr}^{-1}$ for a simulated NPP of 65 $\text{Gtyr}^{-1}$ based on estimates for present day (Galloway et al., 2013; Ciais et al., 2013), and $w_N$ and $w_C$ the standard atomic weights of nitrogen and carbon, respectively.

The losses of nitrogen are given priority over immobilization and plant uptake each time step. Following Meixner and Bales (2002); Thornton et al. (2007); Parida (2011), daily losses by leaching are derived from dissolved nitrogen in soil water and the fraction of soil water lost to rivers per day ($f_{h2o}$) assuming a homogeneous distribution of mineral nitrogen ($N_{smin}$) in the soil volume :

$$F_{leach} = f_s N_{smin} f_{h2o} \tag{16}$$

where $f_s$ is the fraction of mineral nitrogen ($N_{smin}$) in soil solution. $f_{h2o}$ is computed dynamically accounting for evapotranspiration, precipitation, and changes in the soil water storage using a 5 layer soil hydrological scheme (Hagemann and Stacke, 2014)

Following Parida (2011); Goll et al. (2012), daily losses by denitrification are assumed to be at most 0.02% ($k_{denit} = 0.002$ day$^{-1}$) of the soil mineral ($N_{smin}$):

$$F_{denit} = \alpha k_{denit} N_{smin} \tag{17}$$

where $\alpha$ is a JSBACH internal indicator of soil moisture stress [0–1] which is dynamically computed from soil moisture and used to scale biological activity (Raddatz et al., 2007).

## 2.3 Calibration & parametrization of the model

The parametrization of YASSO (version 3.20) and of the nitrogen cycle in JSBACH was not changed and is described in Goll et al. (2012, 2015). The only exception is the re-calibration of losses of nitrogen by leaching to the new hydrological model in JSBACH (Hagemann and Stacke, 2014). This is achieved, following Goll et al. (2012), by tuning the fraction of mineral nitrogen in soil solution ($f_s$) so that the assumption regarding the absence of CNL in the pre-industrial state is met which equals to a negligible (<2%)) effect of nitrogen on global net primary productivity and carbon storage. We tuned the the fraction of soil mineral nitrogen in soil solution to $f_s = 0.1$, which is comparable to fractions used in other models (Wang et al., 2010) as well as in observations (Hedin et al., 1995).

## 2.4 Simulation setup

We force the land surface model JSBACH with half hourly climatic data simulated by the MPI-ESM instead of running JSBACH coupled with the atmospheric and ocean components of the MPI-ESM. Therefore, our simulations, in contrast to simulations of the MPI-ESM, do not account for the feedback between land and atmosphere with respect to the water and energy cycle. However, the resulting inconsistencies between climate and land surface should not change the results of the present study and are anyway partly implicit to the underlying CMIP5 simulations because of the prescribed atmospheric $CO_2$ levels in case of biogeochemical feedbacks (Taylor et al., 2012). For the sake of simplicity, we will refer to the JSBACH simulations driven by the climate from respective ESM simulations, with the respective label of the ESM simulations.

The climatic forcing is derived from MPI-ESM simulations performed for the CMIP5 project (Table 4) (Taylor et al., 2012).

**Table 4.** Simulations performed with JSBACH with and without carbon-nitrogen interactions using climatic forcing from MPI-ESM simulations performed for the CMIP5 project (Taylor et al., 2012).

| Acronym | C-N interactions | climatic forcing | description |
|---|---|---|---|
| C | without | 1pctCO2 | 1% $yr^{-1}$ increase in $CO_2$ (to quadrupling) |
| $C_\beta$ | without | esmFdbkl | Carbon cycle sees piControl $CO_2$ concentration, but radiation sees 1% $yr^{-1}$ rise |
| $C_\gamma$ | without | esmFixClim1 | Radiation sees piControl $CO_2$ concentration, but carbon cycle sees 1% $yr^{-1}$ rise |
| CN | with | 1pctCO2 | 1% $yr^{-1}$ increase in $CO_2$ (to quadrupling) |
| $CN_\beta$ | with | esmFdbkl | Carbon cycle sees piControl $CO_2$ concentration, but radiation sees 1% $yr^{-1}$ rise |
| $CN_\gamma$ | with | esmFixClim1 | Radiation sees piControl $CO_2$ concentration, but carbon cycle sees 1% $yr^{-1}$ rise |

### 2.4.1 Spinup

The concept of $CO_2$ induced nutrient limitation (CNL) assumes that nitrogen effects on the carbon cycle are marginal under pre-industrial conditions. Therefore the cycles of carbon and nitrogen can be equilibrated in a two-step procedure in which the carbon cycle is first brought into equilibrium (less than 1% change in global stocks per decade) using the climatic forcing from the pre-industrial control run (Goll et al., 2012). In a second step, we then initialize the nitrogen pools using the prescribed C:N ratios and the equilibrated carbon stocks as well as extremely high mineral nitrogen pools. The model is run again with the climatic forcing from the pre-industrial control run to equilibrate mineral nitrogen dynamics using the same criterion as for the first step. The resulting length of the simulation is 5.5 kyr and 2.6 kyr for step one and step two, respectively. Atmospheric nitrogen depositions are neglected.

### 2.4.2 1% $CO_2$ increase experiment & climate feedback factors

To analyze the effect of nitrogen limitation on the response of the land carbon cycle to increasing $CO_2$ concentration and climate change, we perform simulations with JSBACH with and without activated nitrogen cycle (Table 4). The simulations are forced with the climatic conditions from a set of 140 yr long CMIP5 simulations with the MPI-ESM in which atmospheric $CO_2$ concentration increases at a rate of 1% $yr^{-1}$ from pre-industrial values until concentration quadruples (Arora et al., 2013). The set of MPI-ESM simulations consist of a simulation where increasing $CO_2$ affects the climate but not the terrestrial biogeochemistry (radiatively coupled), a second simulation where increasing $CO_2$ affects the terrestrial biogeochemistry but not the climate (biogeochemically coupled), and a third simulation where increasing $CO_2$ affects both, climate and biogeochemistry (fully coupled). The biogeochemically-coupled and the radiatively coupled simulations allow us to disentangle the carbon-concentration feedback $\beta_L$ and carbon-climate feedback $\gamma_L$, respectively (Friedlingstein et al., 2006; Arora et al., 2013).

$\beta_L$ is derived from the biogeochemically coupled simulations by dividing the difference in the total land carbon between the first and the last decade by the difference in the atmospheric $CO_2$ concentration of the same periods. $\gamma_L$ is derived from the radiatively coupled simulations by dividing the difference in the total land carbon between the first and the last decade by the difference in global land temperature of the same periods. The MPI-ESM simulations do not include the confounding effects of changes in land use, non-$CO_2$ greenhouse gases, aerosols, etc., and so provide a controlled experiment with which to compare carbon climate interactions in line with the approach by Arora et al. (2013). The model version also does not include dynamic vegetation model and disturbances, such as fire. Natural vegetation cover is prescribed following approach by Pongratz et al. (2008). Cropland and pasture map for 1850 is taken from harmonized landuse dataset by Hurtt et al. (2011).

## 2.5 Analysis

### 2.5.1 Pre-industrial state

We average the model data of last three decades of the spinup simulations to derive the pre-industrial state. Differences between model and observation are given by the subtraction of the observation with the simulation. The fraction of denitrification losses to total losses is computed by dividing the annual flux of denitrification by the sum of the annual fluxes of denitrification and leaching. Simulated and observation loss fractions are compared using Pearson correlation coefficients, RMSE, and Taylor scores (Taylor, 2001).

### 2.5.2 Nitrogen loss pathway data

$\delta$15N data measurements are one of the few sources of spatially extensive data relevant to the nitrogen cycle (Houlton et al., 2015) as one can infer information about the nitrogen pathways. Houlton et al. (2015) derived the fraction of nitrogen loss in gaseous form ($f_{denit}$) based on Amundson et al. (2003) best-fitting multiple regression equation for soil $\delta$15N as a function of mean annual temperature (MAT) and mean annual precipitation (MAP). The data set used to generate this equation consisted of 29 samples, and the coefficient of determination was 0.39. Amundson et al. (2003) remarked that 'pending the availability of more soil $\delta$15N analyses, the present Figure ... represents our best estimate of trends ... in global soil $\delta$15N values' (p. 5). We have updated this analysis in three ways: (a) by including a larger number (659) of soil $\delta$15N samples; (b) by substituting an annually integrated index of temperature-related microbial activity for MAT, and an index of leaching (derived from runoff) for MAP – i.e. using indices more closely related to the governing processes; and (c) by using non-linear regression to fit a statistical model that is explicitly based on the isotopic mass balance equation of (Houlton and Bai, 2009).

The fraction of N loss in gaseous form ($f_{denit}$) was estimated using the principle described by e.g. Houlton and Bai (2009); Bai et al. (2012), but using a process-based statistical model for the relationship between soil $\delta$15N data and environmental predictors fitted to publicly available data on soil $\delta$15N (Patino et al., 2009; Cheng et al., 2009; McCarthy and Pataki, 2010; Fang et al., 2011, 2013; Hilton et al., 2013; Peri et al., 2012; Viani et al., 2011; Sommer et al., 2012; Yi and Yang, 2007). It was assumed that soil $\delta^{15}$N reflects the source (atmospheric) $\delta^{15}$N modified by isotopic discrimination that occurs during leaching (slight) and gaseous losses (much larger). For simplicity, the source $\delta^{15}$N was assumed to be zero and discrimination during

leaching was neglected. Mean annual runoff (q, in [mm]) was estimated from precipitation and potential evapotranspiration following (Zhang et al., 2004), with $\omega = 3$. Following Xu-Ri et al. (2008) we assumed that leaching losses increase to a maximum dependent on soil water capacity, yielding an annual runoff factor f(q):

$$f(q) = \frac{q}{q + W_{max}} \tag{18}$$

with $W_{max} = 150$ mm. Mean monthly soil temperatures ($T_m$, in K) were estimated for 0.25 m depth following Campbell and Norman. We assigned a generic activation energy of $E_a = 55kJmol^{-1}K^{-1}$ (Canion et al., 2014) and summed the monthly index values

$$f_m(T_m) = exp(\frac{E_a}{R_{gas}}(\frac{1}{T_{ref}} - \frac{1}{T_m})) \tag{19}$$

over the 12 months

$$f(T) = \sum_{m=1}^{12} = f_m(T_m) \tag{20}$$

yielding the annual soil temperature factor $f(T)$, where $T_{ref} = 293K$.

     The data were then fitted via $\epsilon$ the gaseous discrimination factor and a constant $k$ by non-linear least-squares regression to the relationship

$$\delta = \delta_0 + \epsilon(1 + k(\frac{f(q)}{f(T)}))^{-1} \tag{21}$$

where $\delta$ is soil $\delta^{15}$N, $\delta_0$ is the $\delta^{15}$N of the N inputs. Assuming the leaching discrimination factor is 0, $f_{denit}$ can be expressed as

$$f_{denit} = \frac{\delta - \delta_0}{\epsilon} \tag{22}$$

from the first principle (Houlton and Bai, 2009). Re-arranging Equation 21 and 22 we get

$$f_{denit} = (1 + k(\frac{f(q)}{f(T)}))^{-1} \tag{23}$$

A spatial map of $f_{denit}$ was derived from the empirical relationship between temperature, runoff and $f_{denit}$ using simulated values of $f(q)$ and $f_m(T_m)$ from JSBACH. Thereby, model biases in climate are accounted for in the data derived $f_{denit}$ which allows a straightforward comparison with simulated $f_{denit}$. In addition, we derived maps of $f_{denit}$ based on monthly grids of observed mean climate from 1961–1990 covering the global land surface at a 10 minute spatial resolution (CRU CL2.0) (New et al., 2002) which are shown in Figure A1.

**Table 5.** Comparison of simulated net primary productivity and biomass carbon as well nitrogen stocks and fluxes for pre-industrial conditions with observation based estimates for 1850 and present day.

| | simulated | observation-based | | |
| | 1850 | 1850 | present day | reference |
| --- | --- | --- | --- | --- |
| NPP ($Gtyr^{-1}$) | 65.1 | – | 50–56 | Ito (2011) |
| biomass carbon (Gt) | 514.7 | – | 470–650 | Saugier and Roy (2001); Ciais et al. (2013) |
| biomass nitrogen (Gt) | 4.6 | – | 3.5 | Schlesinger (1997) |
| mineral nitrogen (Gt) | 1.3 | – | – | |
| total nitrogen (Gt) | 63.6 | – | 60–75 | Galloway et al. (2013) |
| leaching ($Mtyr^{-1}$) | 50.0 | 70 | 13–180 | Galloway et al. (2004, 2013) |
| denitrification ($Mtyr^{-1}$) | 49.2 | – | 43–290 | Galloway et al. (2013) |
| BNF ($Mtyr^{-1}$) | 98.3 | 40–120 | 100–139 | Galloway et al. (2004); Vitousek et al. (2013) |
| mineralisation ($Mtyr^{-1}$) | 717.3 | – | – | |

## 3   Results & discussion

### 3.1   Model evaluation: pre-industrial state

The model simulates nitrogen stocks and fluxes under pre-industrial conditions well within the wide range of the few available observation based estimates (Table 5). Most of the estimates are for present day conditions and thus are not directly comparable due to the human influence on the nitrogen cycle (Galloway et al., 2013).

The organic nitrogen stocks and fluxes are given by the prescribed C:N stoichiometry and the state variables of the carbon cycle and thus are not affected by the changes we introduced here, except for non-lignified litter and fast decomposing soil organic matter which shows in general good agreement with observed C:N ratios for most biomes (see Table A1). Mineralisation of organic nitrogen is the major source of nitrogen for vegetation and the simulated flux is less than existing model based estimates for present day ranging between 980–1030 $Mtyr^{-1}$ (Smith et al., 2014; Zaehle et al., 2010). In models, nitrogen mineralisation is solely a by-product of decomposition of soil organic carbon and we thus attribute the differences between simulated mineralisation to the use of YASSO decomposition model compared to the use of the CENTRUY decomposition model (Smith et al., 2014; Zaehle et al., 2010) as the soil C:N stoichiometries are comparable among models. We refer to the evaluation of the carbon cycle in JSBACH elsewhere (Anav et al., 2013; Goll et al., 2015), as the concept of $CO_2$ induced nutrient limitation prevents an effect of nitrogen on the carbon cycle under pre-industrial $CO_2$ concentrations.

Estimates of global fluxes and stocks of nitrogen are often lacking or associated with large uncertainties, thus a detailed analysis of the simulated nitrogen cycle is hampered (Zaehle, 2013). However, recent advances in the use of $\delta^{15}N$ data (Houlton et al., 2015), which are one of the few sources of spatially extensive data relevant to the nitrogen cycle, allow the evaluation of the respective importance of nitrogen loss pathways in space. Due to the different environmental controls of the loss pathways,

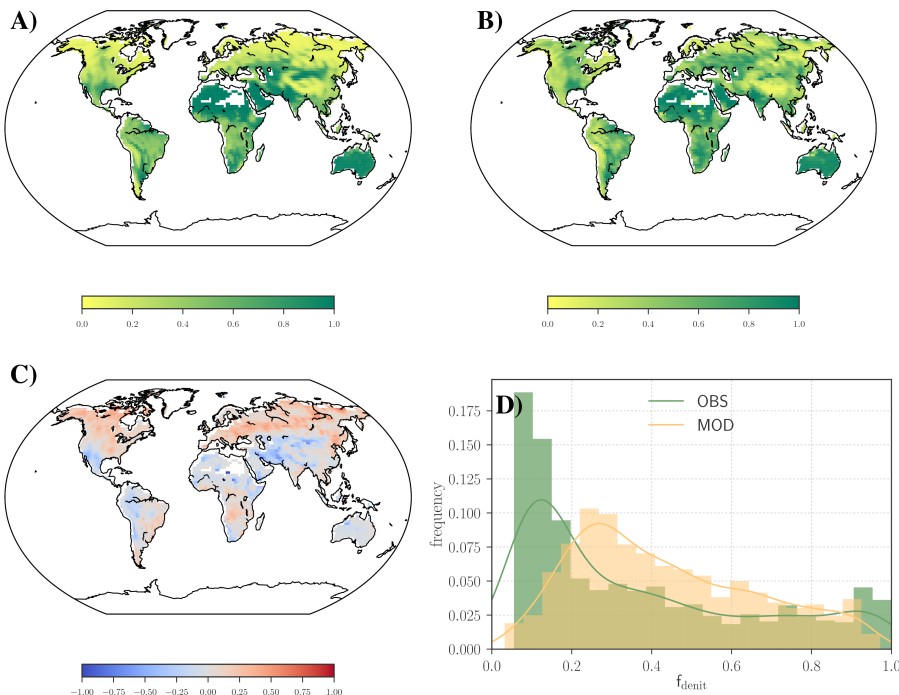

**Figure 2.** Comparison of simulated and reconstructed nitrogen lost by denitrification as a fraction of total nitrogen losses ($f_{denit}$). Shown are $f_{denit}$ reconstructed from $\delta^{15}N$ measurements and simulated climate (A), $f_{denit}$ as simulated (B), the difference between simulated and reconstructed $f_{denit}$ (C), as well as the frequency distribution of simulated (yellow) and reconstructed (green) $f_{denit}$ (D).

which are on first order represented in the model, we can test the underlying assumptions by comparing the simulated fraction of denitrification losses to total nitrogen losses ($f_{denit}$) to $f_{denit}$ reconstructed from $\delta^{15}$N data. However, this comparison does not allow to draw any conclusion about the magnitude of total losses.

The reconstructed $f_{denit}$ maps (Figure (A1&2) presented here are generally similar to those presented by Houlton et al.
5 (2015), with high fractions (ca 80%) in the tropics and mid-latitude deserts, a strong gradient of decreasing fractions with decreasing temperature towards high altitudes and latitudes, and values in the range 0-20% reached in cold, wet climates in the north. For a detailed discussion of differences see SI.

In comparison with the reconstructed fractional gaseous loss from simulated climate (Figure 2a), we find that the model is in good agreement (Pearson R=0.76, RMSE=0.2, Taylor score=0.83). The model underestimates high values of $f_{denit}$ and
10 overestimate low values (Figure A2). In regions with cold winter temperatures where denitrification losses are small the model overestimates denitrifcation losses (Figure 2c). These model biases likely derive from the simplistic representation of denitrifi-cation as a function of soil moisture and substrate availability, which omits effects of temperature (Butterbach-Bahl et al., 2013). Additionally, other omitted factors like oxygen concentration, soil pH, mineralogy, and transport processes (Butterbach-Bahl et al., 2013) might contribute to the bias.

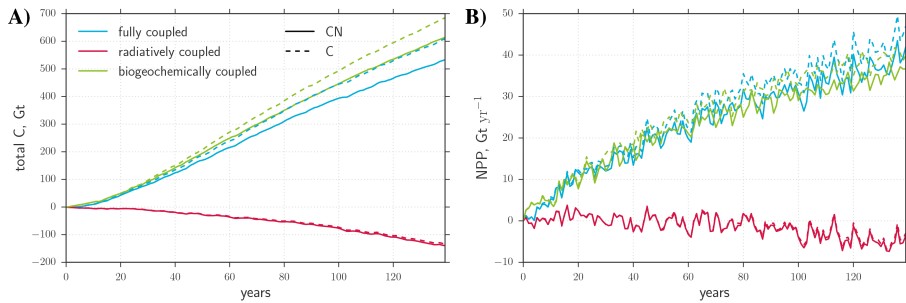

**Figure 3.** Changes in total land carbon (A) and global net primary productivity (B) and in the set of 1% $CO_2$ increase simulations with (solid line) and without (dashed line) carbon-nitrogen interactions.

**Table 6.** Simulated response ratios of gross and net primary productivity to elevated $CO_2$ in comparison with observation based estimates.

| Response ratio | simulated | observed | reference |
|---|---|---|---|
| $GPP_{396}/GPP_{295}$ | $1.23\pm0.03$ | 1.25 | Ehlers et al. (2015) |
| $NPP_{550}/NPP_{370}$ | $1.16\pm0.03$ | $1.23\pm0.02$ | Norby et al. (2005) |

### 3.2 Changes in the land carbon cycle in the 1% $CO_2$ increase simulations

JSBACH simulates a strong increase in net plant productivity (NPP) due to increasing $CO_2$ from pre-industrial level to $4 \times$ pre-industrial level (Figure 3). The simulated increase in NPP of 16.0% for a rise in atmospheric $CO_2$ from 370 ppm to 550 ppm is somewhat lower than the estimated increase of 23% from 4 free air carbon dioxide enrichment (FACE) experiments

(Norby et al., 2005) (Table 6). A lower increase than in the FACE experiment can be expected as the long-term effect of elevated $CO_2$ is likely to be less than the one derived from short duration FACE experiments (Norby et al., 2010) on early successional forests (Norby et al., 2015). The simulated increase in GPP of 23.1% for an increase in atmospheric $CO_2$ concentration from the level of the year 1900 to 2013 is close to the 25% increase for the same increase in $CO_2$ estimated from intramolecular isotope distributions (isotopomers), a methodology for detecting shifts in plant carbon metabolism over long times (Ehlers

et al., 2015).

    The increase in NPP translates to an increase in carbon storage of approximately 600 Gt by end of the biogeochemically-coupled simulation (Figure 3). Climate change, in particular increasing temperature, overall has a slightly negative effect on global NPP: the carbon losses by autotrophic respiration in low latitudes outweigh the increases in NPP in temperature limited ecosystems. Globally, warming stimulates the decomposition of soil organic matter (not shown) which leads to a smaller

increase in carbon storage in the fully-coupled simulation compared to the biogeochemically-coupled simulation, and even a reduction in carbon storage in the radiatively-coupled simulations. The effect of $CO_2$ and climate change on land carbon storage is much less pronounced in the recent version of JSBACH than in the CMIP5 version and the responses of the new

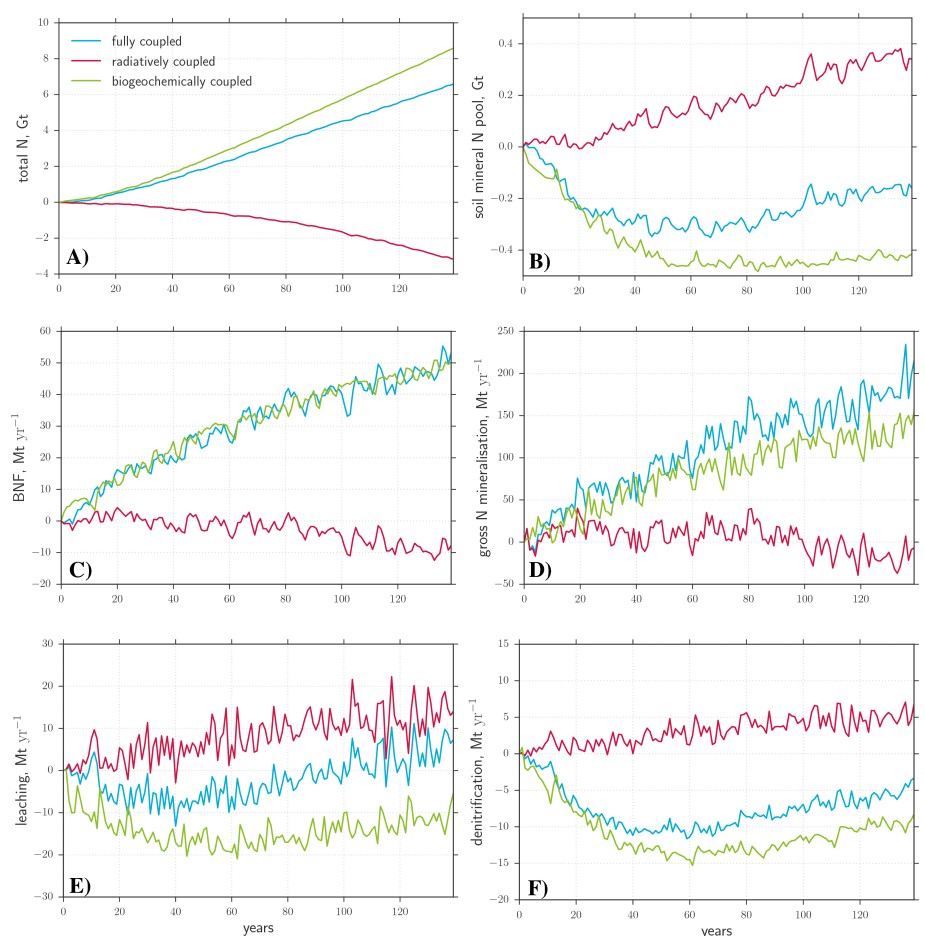

**Figure 4.** Changes in the nitrogen cycling in the set of 1% $CO_2$ increase simulations with carbon-nitrogen interactions: (A) total nitrogen, (B) mineral N, (C) biological nitrogen fixation, (D) gross mineralisation, (E) leaching, and (F) denitrification.

version lie well within the range of CMIP5 models (Arora et al., 2013). The more moderate response can mainly be attributed to the recent improvement in respect to the carbon cycle and are discussed in detail later.

### 3.3 Changes in the land nitrogen cycle in the 1% $CO_2$ increase simulations

Increasing atmospheric $CO_2$ concentration leads to the accumulation of nitrogen in the terrestrial system (Figure 4a), due to elevated inputs by biological nitrogen fixation (BNF) (Figure 4c) in combination with reduced losses by leaching and denitrification (Figure 4e,f). The increasing primary productivity and the subsequent incorporation of soil mineral nitrogen are the main drivers behind the accumulation. Increasing NPP (Figure 3) directly stimulates the demand-driven process of BNF and therefore BNF rates increase nearly as strong as NPP (50% compared to 59% by end of the simulations). The decline

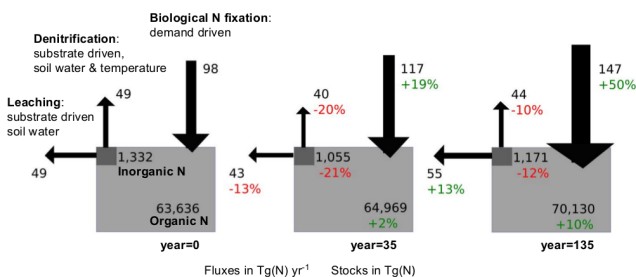

**Figure 5.** The terrestrial nitrogen balance during the fully "coupled" simulation. Shown are the average fluxes and stocks for 10 yr time period for initial conditions (year=0; 284 ppm), fourth decade (year=35; 400ppm) and the last decade (year=135; $4 \times 284$ ppm).

in soil mineral nitrogen (Figure 4b) due to the incorporation of nitrogen in accumulating biomass leads to reduced losses by leaching and denitrification.

An increase in the nitrogen stock by 8.5% was found in a 15yr ecosystem scale $CO_2$ enrichment experiment (Shi et al., 2016), which is more pronounced than the simulated increase in the nitrogen stock of 3.2% for a comparable increase in $CO_2$ (year 29–69; 369–551ppm). A strong stimulation of BNF by 44%, with a strong decline in leaching by 42% and no significant changes in denitrification, mineralisation and soil organic nitrogen were found in a compilation of $CO_2$ enrichment experiments (Liang et al., 2016). However the representativeness of these findings was questioned recently (Rütting, 2016). In addition to that, $CO_2$ was increased abruptly in $CO_2$ enrichment experiments while it increased gradually in our simulations. As the different nitrogen processes have different response patterns, t hey are likely to react differently to an abrupt than to a gradual increase in $CO_2$. Although the relative contributions of reduced losses and increased inputs to an accumulation remain somewhat elusive due to methodological biases (Rütting, 2016) and limited data, an accumulation of nitrogen under elevated $CO_2$ is a plausible scenario.

We find that the processes governing the nitrogen balance operate on different time scales (Figure 5). The mineral nitrogen stocks decline from 1.33 Gt to 1.06 Gt during the first 35 yr and thereby reduce the substrate driven nitrogen losses (Figure 5) from 98 $\mathrm{Mtyr^{-1}}$ to 83 $\mathrm{Mtyr^{-1}}$. However, losses by leaching start to increase afterwards and are higher by the end of the simulation than at the start. While the reduction in losses and gains in inputs contribute to equally parts to the accumulation in the first decades of the simulation, the stimulated BNF dominates the accumulation in later (Figure 5). This highlights the importance of long term manipulation experiments for improving our understanding about the long term effects of increasing $CO_2$ on the terrestrial biosphere.

We find that the effect of increasing $CO_2$ and climate change on the nitrogen balance differ. Elevated $CO_2$ alone leads to a shift from inorganic nitrogen to organic nitrogen (Figure 4a&b), whereas climate change is dampening this shift as warming stimulates the decomposition of organic nitrogen (Figure 4d) and thereby slow down the progressive immobilization of mineral nitrogen into biomass and soil organic matter. Climate change alone leads to a loss of nitrogen from the system (Figure4a): The enhanced mineralisation of organic nitrogen due to warming leads to increased losses of nitrogen via leaching and denitrification. We further found that changes in the water cycle due to climate change are increasing losses by leaching, while

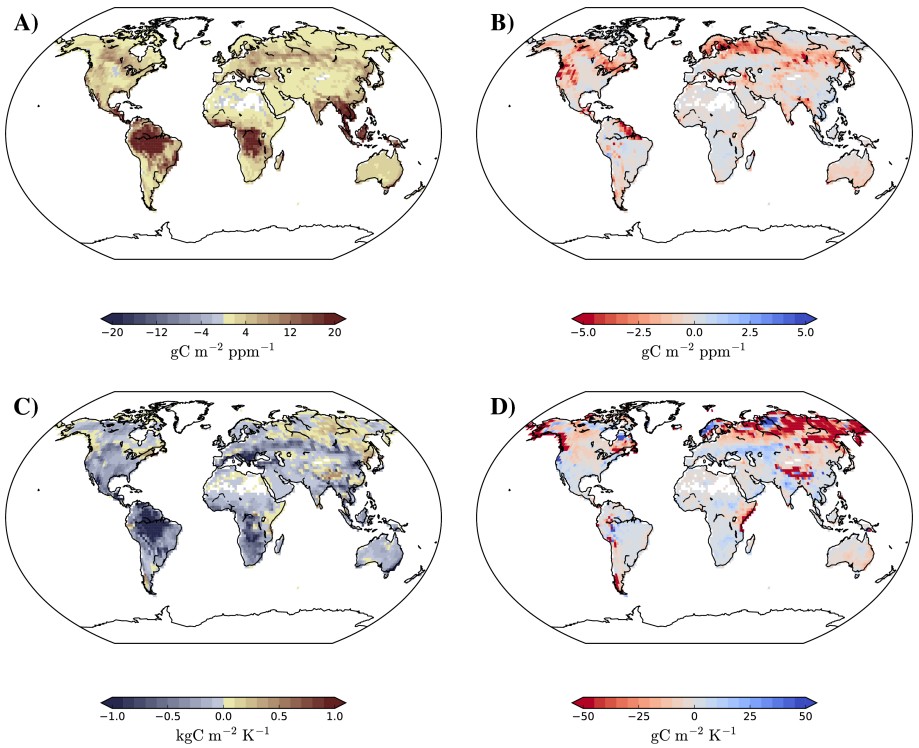

**Figure 6.** Spatial map of the carbon concentration feedback $\beta_L$ (A) and the carbon climate feedback $\gamma_L$ (B) (of the simulations with carbon-nitrogen interactions) as well as the effect of the nitrogen cycling on the respective feedbacks (C,D).

denitrification follows primarily the changes in substrate availability despite the influence of soil moisture in Eq. 17 (Figure 5). Overall, we find that total nitrogen losses are intensified relative to the substrate availability at the end compared to the start of the simulations (Figure 5).

5   The simulated increase in tightness of the nitrogen cycle as mineral nitrogen stocks deplete is in line with the substrate-based mechanisms proposed based on recent compilation of ecosystem nitrogen addition experiments (Niu et al., 2016), in which the mineral nitrogen exerts a major control on the mineral nitrogen consuming processes. However, the respective observed response patterns of ecosystem nitrogen processes remain to a large degree unknown and are represented in a strongly simplified way in JSBACH . In general, we find that the effect of nitrogen availability on carbon storage is rather moderate in all simulation (Figure 3) due to the adjustments of the nitrogen balance to changes in the carbon cycle (Figure 5).

10   **3.4   The effect of nitrogen on the carbon feedbacks**

We quantify the strengths of the climate carbon-feedback ($\gamma_L$) and the carbon-concentration feedback ($\beta_L$) from the radiatively "coupled" and biogeochemically "coupled" simulations, respectively (Figure 6a,c). Both land feedbacks, with a global $\beta_L$ of 0.61 Pg(C)ppm$^{-1}$ and a global $\gamma_L$ of -27.5 Pg(C)$^{\circ}$C$^{-1}$, are 34% and 53% smaller than the multi-model averages of the

**Table 7.** Carbon cycle feedbacks in simulations with JSBACH compared to results from CMIP5.

|  | Land carbon-concentration feedback $[Pg\,ppm^{-1}]$ | Land carbon-climate feedback $[Pg\,^{\circ}C^{-1}]$ | reference |
|---|---|---|---|
| CMIP5 MPI-ESM-LR | 1.46 | -83.2 | Arora et al. (2013) |
| CMIP5 multi model mean | 0.92±0.44 | -58.5±28.5 | Arora et al. (2013) |
| JSBACH C | 0.74 | -26.2 | this study |
| JSBACH CN | 0.61 | -27.5 | this study |

CMIP5 models, despite that the CMIP5 version of JSBACH simulated global $\beta_L$ and $\gamma_L$ which are 59% and 42% times larger than the average of CMIP5 models (Table 7).

In CMIP5 the two ESMs with nitrogen limitation (which shared the same terrestrial biosphere component) had feedback strengths 70–75% lower than averaged across models (Arora et al., 2013), suggesting a prominent role of nitrogen in dampening both carbon cycle feedbacks. Here, we find that the dampened response is primarily related to the modifications of the soil and litter carbon decomposition module, rather than to the inclusion of the nitrogen cycle. The global $\beta_L$ in the simulation without nitrogen cycle is only 10% larger than in the simulation with nitrogen, while there is hardly any difference in global $\gamma_L$ between simulation, as the small positive and negative differences cancel out (Figure 6). The contrasting findings regarding the effect of nitrogen on the land carbon feedbacks illustrates the need of a multitude of carbon-nitrogen models to draw general conclusions.

The large difference between the CMIP5 version of JSBACH and the recent version described here can primarily be attributed to the new decomposition model. The drastically reduced $\gamma_L$ is primarily caused by the smaller initial soil carbon stock, as well as by the long-term acclimation of decomposition to warming due to substrate depletion in YASSO (Goll et al., 2015). The importance of the initial soil carbon stock for the carbon losses due to warming was illustrated by Todd-Brown et al. (2014). The lower $\beta_L$ can be attributed to a much lower fraction of biomass which is converted into stable soil organic matter. Therefore, the increasing productivity translates to a much smaller increase in carbon storage. JSBACH does not account for the stimulation of decomposition of recalcitrant carbon under elevated $CO_2$ due to increases in labile organic matter (from of priming effect) observed in lab incubation experiments (Kuzyakov et al., 2000), which could potentially alter the response of soil carbon to increasing $CO_2$, but for which there is no evidence of its relevance on multi-decadal time scales (Cardinael et al., 2015).

A dampening effect of the nitrogen cycle on the response of the terrestrial carbon cycle to climate change and increasing $CO_2$ is in line with the majority of carbon-nitrogen model studies (Ciais et al., 2013; Zaehle, 2013), but not all (Esser et al., 2011; Warlind et al., 2014). The mechanism by which nitrogen can dampen $\beta_L$ is via an increasing decoupling of gross primary productivity from biomass accumulation under increasing $CO_2$ concentration: the incorporation of nitrogen into biomass reduces the mineral nitrogen availability (Luo et al., 2004; Liang et al., 2016) which negatively affects growth (Norby et al., 2010) and increases root respiration (Vicca et al., 2012; Mccormack et al., 2015). The dampening of $\gamma_L$ is mainly via an en-

hanced nitrogen mineralisation in cold regions due to warming (Zaehle, 2013; Warlind et al., 2014; Koven et al., 2015), which cannot be fully captured by JSBACH due to assumption of $CO_2$-induced nutrient limitation, and therefore the model is prone to underestimate the effect of nitrogen on $\gamma_L$.

## 3.5 Model limitations and future development directions

The current understanding of processes governing the terrestrial nitrogen balance is still rather limited (Zaehle, 2013), and several processes which might be of importance, in particular stand dynamics (Warlind et al., 2014) which can potentially alter biomass turnover (Brienen et al., 2015), interactions between plants and microbes which can stimulate nitrogen scarcity by non-altruistic symbioses (Franklin et al., 2014), plasticity in stoichiometry and leaf nutrient recycling (Zaehle et al., 2014; Meyerholt and Zaehle, 2015), and the availability of other nutrients (Goll et al., 2012) are not represented in JSBACH. In
addition, the loss of organic matter in general due to erosion, although potentially of importance (Lal , 2002), is not yet represented in global land surface models, but development are underway (Naipal et al., 2015, 2016).

Due to the concept of $CO_2$-induced nutrient limitation in JSBACH the nitrogen cycle serves primarily as an additional constraint on the carbon uptake. The advantage of the approach is its low complexity and avoidance of assumptions about the initial state of nutrient limitation thereby taking into account (1) the lack of data regarding the nitrogen cycle (Zaehle, 2013)
as well as (2) the large uncertainty about the nutrient constraint on plant productivity (Letters et al., 2007; Zaehle, 2013). The shortcomings of this approach are that it limits the applicability of the model to carbon cycle projections for scenarios of increasing atmospheric $CO_2$ and that it cannot capture any stimulation of the plant productivity due to changes in nitrogen availability itself: In addition to direct increase in nitrogen availability by nitrogen deposition and fertilization, a stimulation of plant productivity can occur due to reduced losses of nitrogen by pathways which are not under control of biota, like fire,
leaching, or erosion (Thomas et al., 2015). As a result, the model might underestimate the importance of nitrogen cycling for carbon uptake under elevated $CO_2$.

Regarding the processes resolved in JSBACH, the extent of BNF increase has to be regarded as highly uncertain, despite its agreement with short-term experiments (Liang et al., 2016): the formulation of BNF used here is based on an empirical correlation between evapotranspiration and BNF and therefore the rate at which BNF rates increase strictly follows the increase
in productivity whereas in reality the different processes leading to changes in BNF on ecosystem scales operate on a different time scale: The control of plants on their symbiotic partners via glucose export, and in case of nodules via oxygen regulation, result in changes in BNF from hours to months. On longer time scales, the composition of the ecosystem, namely the fraction of BNF associated species, affects nitrogen inputs to the system. While for tropical ecosystems there is evidence that any governing mechanism(s) ought to operate at a synoptic scale (Hedin et al., 2009), higher latitude system might experience
longer lag times. Additionally, other nutrients, like phosphorus or molybdenum, might slow down or reduce the potential of BNF to increase (Vitousek et al., 2013). BNF models which better resolve the governing mechanisms, for example (Gerber et al., 2010; Fisher et al., 2012), should be incorporated into ESMs to increase the reliability of the simulated pace of changes in BNF (Meyerholt et al., 2016).

Models which simulate simultaneous competition for soil nitrogen substrates by multiple processes match the observed patterns of nitrogen losses better than models like JSBACH which are based on sequential competition (Niu et al., 2016). Here we find that despite the sequential competition, the simulated behavior is in general agreement with the dynamics of substrate-based mechanisms derived from manipulation experiments (Niu et al., 2016) and the spatial variability in the respective loss pathways is to a large degree in line with $\delta^{15}$N-derived patterns, despite the low performance of another sequential competition model (Houlton et al., 2015; Zhu and Riley, 2015)

The high-latitude permafrost processes are not represented here but were shown to be of importance for the effect on warming on carbon and nitrogen losses. Permafrost regions store about 1,000 Gt C within the upper few meters of soil (Hugelius et al., 2014). The thawing of permafrost and deepening of the active layer in response to global warming can potentially lead to a much stronger climate carbon feedback (Schneider Von Deimling et al., 2012; Schuur et al., 2015). The recent study of Koven et al. (2015) with the CLM model showed these carbon losses in high latitudes can be partly offset by increased nitrogen mineralisation, and in turn productivity and input to the soils.

Finally, as advocated by for example in Prentice et al. (2014); Medlyn et al. (2015), the stringent use of observational data set to evaluate the present day state of ecosystems as well as their response to manipulations must drive and guide new model developments whenever possible. With respect to JSBACH and other land surface models, the use of observation-derived climatology instead of the ESM climate as well as the use of site-specific simulations to allow a straightforward comparison to manipulation experiments is a research priority to increase the model reliability (Luo et al., 2012). In this study, the use of the ESM climatology is justified by a focus on the feedback analysis in the framework of idealized simulations as suggested in the climate-carbon cycle model intercomparison project C4MIP (Anav et al., 2013; Jones et al., 2016). For further evaluation of the nitrogen limitation, however the preferable setup includes site-level simulations driven by observation-derived climatology (Zaehle et al., 2014).

# 4 Conclusions

The simulated response of primary productivity to increasing $CO_2$, simulated litter stoichiometries, as well as the simulated spatial variability in nitrogen loss pathways are in good agreement with observation based estimates. Here we show that a simple representation of mineral nitrogen dynamics can achieve a high agreement with observation in respect to nitrogen loss pathways. Further refinements of denitrification should address the relationship between denitrification and low soil moisture availability and as well as introduce a temperature scaling function.

The effect of nitrogen cycling on the land carbon uptake in idealized simulations with JSBACH is globally minor, but not negligible. In particular, the carbon-concentration feedback is affected by mineral nitrogen availability, but the extent is moderate compared to earlier studies (Arora et al., 2013; Zaehle, 2013). During the first decades of the simulations, nitrogen limitation is circumpassed by a strong initial decline in loss terms in combination with increases in biological nitrogen fixation. Afterwards progressive increases in biological nitrogen fixation drive the accumulation of nitrogen in ecosystems. On top of that, warming enhances mineralisation and counteracts the immobilization of nitrogen in biomass. Our study is in line with the

majority of carbon dioxide enrichment studies (Liang et al., 2016), showing that progressive nitrogen limitation under elevated carbon dioxide concentrations is less likely to occur than originally suggested (Luo et al., 2004).

The timescale and the extent to which the nitrogen cycle adjusts to increasing carbon dioxide and changing climate depend on the response of the input and loss processes of nitrogen as well as turnover of organic nitrogen. Here, we illustrate that the processes countering nitrogen scarcity operate on different time scales and have different trajectories due to differences in the respective environmental drivers, which indicates a picture more complicated than drawn from environmental response functions (Niu et al., 2016; Liang et al., 2016). It is difficult to assess to what extent the timescales in our experiments are realistic, as timescale on which these processes operate is well beyond the typical duration of manipulation experiments.

Our results suggest that nitrogen limitation of land carbon uptake of natural ecosystems could be temporally restricted, being the result of the inertia of the balancing processes (Altabet et al., 1995; Hedin et al., 2009). Ultimately, other nutrients like phosphorus which sources are depleted over time, are likely to dominate the long-term capacity of carbon storage.

## 5   Code availability

The JSBACH model version 3.10 used here includes the soil module YASSO and nitrogen components. The model version corresponds to the revision 8691 from the 19th July 2016 in the Apache version control system (SVN) of the Max Planck Institute for Meteorology (https://svn.zmaw.de/svn/cosmos/branches/mpiesm-landveg). This version will be used in the CMIP6 simulations, where other components (landuse, dynamic vegetation, fire) will be included as well. The source code of the CMIP6 version of JSBACH as a part of MPI-ESM will be available in 2017 under the MPI-M Software License Agreement obtained at http://www.mpimet.mpg.de/en/science/models/license/. In meantime, please contact Thomas Raddatz (thomas.raddatz@mpimet.mpg.de) for the code of the JSBACH if you plan an application of the model and envisage longer-term scientific collaboration.

## 6   Data availability

Primary data and scripts used in the analysis and other supplementary information that may be useful in reproducing the author's work are archived by the Max Planck Institute for Meteorology and can be obtained by contacting publications@mpimet.mpg.de.

## Appendix A:  The nitrogen limitation factor

The nitrogen limitation factor, $f_{limit}^N$, is calculated based on a supply and demand approach (Parida, 2011; Goll et al., 2012). In a first step, potential carbon fluxes are computed from which the gross mineralisation, immobilization ($D_{micr}$) and plant uptake of mineral nitrogen ($D_{veg}$) is diagnosed. In a second step, all fluxes consuming nitrogen (donor compartment has a higher C:N ratio than the receiving pool as well as plant uptake) are down-regulated in case nitrogen demand cannot be met by the nitrogen supply. Hereby, a common scalar ($f_{limit}^N$) is used thereby no assumption about the relative competitive strengths

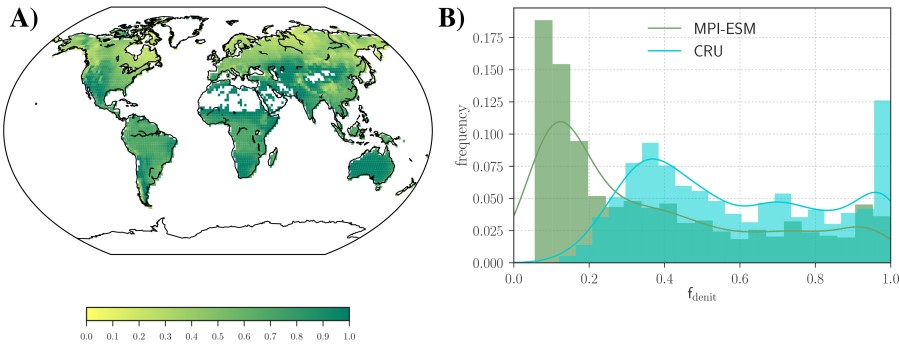

**Figure A1.** Reconstructed fractions of nitrogen lost by denitrification relative to total losses ($f_{denit}$). Shown are loss fractions reconstructed from observational data on $\delta^{15}N$ and observed climatology (A) and the frequency distribution of reconstructed $f_{denit}$ from observed (turquoise) and simulated (green) climatology, respectively (B).

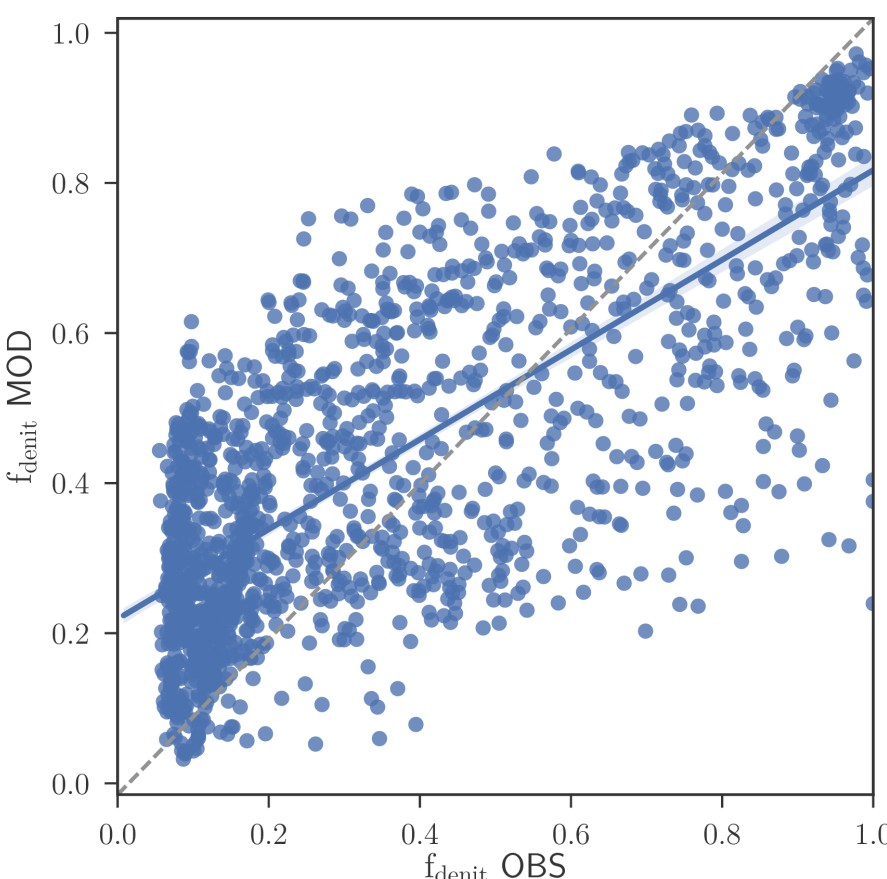

**Figure A2.** Scatter plot of simulated and reconstructed denitrifcation fractions ($f_{denit}$) derived from simulated climatology.

**Table A1.** Carbon to nitrogen mass ratio $[\text{g}(\text{C})\text{g}^{-1}(\text{N})]$ of non-lignified litter compared to observations of foliage litter from Cornwell et al. (2008); Brovkin et al. (2012)

| PFT | simulated | observed |
|---|---|---|
| tropical broadleaved evergreen trees | 53.5 | 55.9 |
| tropical broadleaved deciduous trees | 55.1 | 29.4 |
| extra-tropical evergreen trees | 50.4 | 68.3 |
| extra-tropical deciduous trees | 54.4 | 55.9 |
| C3 perennial grass | 54.5 | 47.6 |
| C4 perennial grass | 53.4 | 54.1 |

of microbial and plant consumption has to be made.

$$
f_{limit}^{N} = \begin{cases} \frac{[\frac{dN_{smin}}{dt}]^{max}}{D_{micr}+D_{veg}} & \text{for } (D_{micr} + D_{veg}) > [\frac{dN_{smin}}{dt}]^{max} \\ 1 & \text{otherwise} \end{cases} \tag{A1}
$$

where the term in square bracket is the maximum rate at which the soil mineral nitrogen pool can supply nitrogen. Note that in the discretized formulation the mineral nitrogen pool can at most be depleted during a single model time step ($\Delta t$). We thus
set this maximum rate to $\frac{dN_{smin}}{\Delta t}$.

## Appendix B: Evaluation of dynamically computed C:N ratios

The only ecosystem compartment in JSBACH which has a flexible stoichiometry is non-lignified litter and fast-decomposing organic matter. The simulated carbon to nitrogen ratios of this compartment for the six plant functional types in JSBACH are in rather good agreement with observations of foliage litter from the ART-DECO database (Table A1), except for tropical
broadleaved deciduous trees and extra-tropical evergreen trees. The reason for the overestimation of nitrogen content in litter from extratropical evergreen trees is the global parametrization of leaf stoichiometry applied in JSBACH which does not capture the lower leaf nitrogen concentration in needle-leaved trees compared to broad-leaved trees (KATTGE et al., 2011). The data for tropical species is very scarce and the variability among species is large, which hamper the interpretation of the mismatch between model and observation for the tropical broadleaved deciduous trees.

**Appendix C: Consistency of nitrogen loss pathways with earlier estimates**

The reconstructed $f_{denit}$ map from observed climatology (Figure (A1) is generally similar to one presented by Houlton et al. (2015), with high fractions (ca 80%) in the tropics and mid-latitude deserts, a strong gradient of decreasing fractions with decreasing temperature towards high altitudes and latitudes, and values in the range 0-20% reached in cold, wet climates in the north. However, some differences are apparent, most obviously connected with the use of mean annual temperature (MAT) by

Houlton et al. (2015) to index microbial activity. MAT becomes extremely low in Eurasia towards the northeast, for example, and accordingly, Houlton et al. (2015) estimates of the denitrification fraction become very low there. Craine et al. (2015) noted that climates with very low MAT (including sites in NE Siberia) showed anomalous values of soil $\delta^{15}$N, more similar to those of warmer climates. Our approach takes account of this by the use of an index that is much more responsive to the warm summers than to the extreme cold winters found in hypercontinental climates.

When simulated climatology is used to upscale the empirical relationship between temperature, runoff and soil $\delta^{15}$N, the influence of biases in simulated climatology on $f_{denit}$ become apparent. The overestimation of precipitation and subsequently runoff of about 20% in MPI-ESM (Weedon et al., 2011; Hagemann et al., 2013) leads to a pronounced peak in the histogram of $f_{denit}$ at about 0.1-0.2 (Figure A1), which is mostly in the mid and high latitudes regions in northern hemisphere.

*Author contributions.* The study was led by Daniel S. Goll who developed the model concept and implemented it into JSBACH. Victor Brovkin, Alexander J. Winkler and Thomas Raddatz designed and performed the model experiment. Alexander J. Winkler analysed and visualized the data. Ning Dong and Colin Prentice performed the estimation of the nitrogen loss pathways. Philippe Ciais contributed to the analysis and evaluation of the model. The draft of the manuscript was written by Daniel S. Goll with all authors contributing to its final form.

*Competing interests.* The authors declare that they have no conflict of interest.

*Acknowledgements.* The authors sincerely thank the editor Dr. Sato and the two anonymous referees for their comments which substantially improved the quality of this work. This work was initiated when DSG was funded through the DFG Cluster of Excellence CLiSAP (EXC 177/2), now he is funded by the "IMBALANCE-P" project of the European Research Council (ERC-2013-SyG-610028). This work contributes to the H2020 project CRESCENDO, which receives funding from the European Union's Horizon 2020 research and innovation programme under grant agreement no. 641816. DN was supported by an international Macquarie University Research Scholarship and by the Ecosystem Modelling and Scaling Infrastructure (eMAST, http://www.emast.org.au) facility of the Terrestrial Ecosystem Research Network. This work is a contribution to the AXA Chair Programme in Biosphere and Climate Impacts and the Imperial College initiative on Grand Challenges in Ecosystems and the Environment (ICP). We thank Christian Reick and Daniela Kracher for discussing the nitrogen model implementation, and Christian Wirth and Hans Cornelissen for discussions of woody litter decomposition. We thank Thomas Kleinen for reviewing the manuscript prior submission. We acknowledge the World Climate Research Programme's Working Group on Coupled Modelling, which is responsible for CMIP, and we thank the climate modeling group at the Max Planck Insitute for Meteorology producing and making available their model output. For CMIP the U.S. Department of Energy's Program for Climate Model Diagnosis and Intercomparson provides coordinating support and led development of software infrastructure in partnership with the Global Organization for Earth System Science Portals.

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
