# Peer review of "Carbon-nitrogen interactions in idealized simulations with JSBACH (version 3.10)"

_Geoscientific Model Development, 2016_

## Referee Comment (RC1) · Anonymous Referee #1 · 6 Feb 2017

The paper of Goll et al. focuses on the carbon-nitrogen interactions in the new version of JSBACH with updated soil organic matter decomposition model and N component. As nutrient limitation plays an important role in land carbon cycles, the work of Goll et al. is very important towards better predicting of future carbon dynamics. The work is novel in making use of nitrogen isotope data to evaluate process based N simulations. The study is generally well conducted and sufficient for recording model behavior, but I have several concerns or questions listed below.

1. To what extent the C-N interactions produced from this paper are reliable?

There is a large uncertainty in N cycle, which makes the C-N interactions difficult to constrain. The N limitation on carbon cycle, or C-N interaction strength, is based on assumptions of $CO_2$ induced nutrient limitation (CNL) from this paper. The assumption of marginal nitrogen effect on pre-industrial C cycle is debatable. Although the model can be parameterized based on preindustrial conditions (which may have large uncertainties) that have already taking into account of the N effect, it may misrepresent important mechanisms that regulate long term N effects on C. For example, losses of plant uncontrollable nitrogen, such as through fire, erosion, dissolved organic matter, constrains long term N availability and therefore N impacts on C [Thomas et al., 2015]. Plant uncontrollable nitrogen loss pathways are not represented in the current version of JSBACH. Therefore nitrogen limitation cannot be maintained with strong biological controls on N losses and inputs in JSBACH in the long run, but that does not mean there is no N limitation in the long run in a real world. It may capture transient $CO_2$ responses. As mentioned by the authors, it may misrepresent climate response and potentially the C-N interactions and other aspects that affect C-N interactions. I suggest the authors be cautious about reaching a conclusion about getting the decomposition of soil carbon right first before incorporating C-N interactions as the evaluation should be based on the right representation of C-N interactions and compared to the "true" observation.

2. How does reproducing the relative fraction of nitrogen loss pathways affect land carbon? Or how does C-N models benefit from an accurate representation of the relative N loss pathways.

It seems to me the focus of this paper is on C-N interaction. Does accurate representation of the relative N loss pathway (leaching vs. gaseous) help in correcting C-N interactions? It is possible to have a correct leaching: gaseous loss ratio while have a wrong simulation of leaching loss or gaseous loss. As the ratio can be tuned through parameters, such as the fraction of soil water lost to rivers per day, the fraction of mineral nitrogen in soil solution, and fdenit estimation from 15N relies strongly on climatic conditions, a reasonable representation of the spatial pattern of fdenit does not necessarily mean a good simulation of mineral N and N limitation on biological activities. It makes the 15N based evaluation more valuable if the authors can clarify the merits of

such evaluation for the general N cycling and C-N interaction.

3. Model description is not very clear and is confusing in some parts.

As this paper focus on how N affects land carbon simulation, it is better to let readers know how N limitation regulates photosynthesis (GPP or NPP) and organic matter decomposition which are two key nexus points in C-N interactions. As the model descriptions combine terms from the YASSO model, the old JSBACH model and the updated JSBACH model, it is difficult to follow especially when the structures of these models are not the same. I suggest reworking on model description. More detailed suggestions are available in Minor Points.

Minor Points:

1. Is it appropriate to have many citations in abstract?

2. Page 1, line 10, the reference of Shi et al., 2015 is not relevant. Do you mean Shi Z, Yang Y, Luo Y, Zhou X, Weng E, Finzi A. 2015. Inverse analysis of coupled carbon-nitrogen cycles against multiple datasets at ambient and elevated CO2. Journal of Plant Ecology, doi:10.1093/jpe/rtv059

3. P4,L5, equation 1, line 9, is there a H component in the matrix equation?

4. P5, lines 2-3 "lignified litter and fast decomposing organic matter" is confusing as no "fast decomposing organic matter" is mentioned in the carbon part.

5. P7, equation 10, where is the nitrogen flux from your la class (non-lignified &fast decomposing organic matter)? The third term, the lignified flux is not clear. Why do you have (rw-rlw)*F for lignified flux while only have rla*F for non-lignified flux in equation 7? The description from Lines 20-21 is not clear. Why do you differentiate N-to-C ratio of lignified litter and biomass?

6. P9, L10, "cchange" to "change "

7. Section 2.5.2 Nitrogen loss pathway data. I may have missed some part, but the

description on how to estimate fdenit is not very clear. You fitted data to equation 19. So in equation 19, what are known and what are need to be estimated? If k is the only factor need to be estimated, what is the purpose to estimate k as equation 20 based on which to estimate fdenit does not need k

8. P12, L3-5 No compiled mineral N in Table 5 is available for comparison and indicates simulated mineral nitrogen stock is within the range of estimates. There is no available data in Table 5 to compare denitrification between simulated 1850 vs. observation-based 1850. Comparing between simulated value at 1850 with present is not appropriate as nitrogen cycle is altered strongly by anthropogenic activities since the industrialization.

9. P12,L6, Is nitrogen in la (non-lignified litter & fast decomposing organic matter) part of the organic nitrogen stocks? Equation 7 says it is not prescribed based on C:N.

10. P13, if climatic forcing is the reason for mismatch, is it feasible to calculate fdenit (the isotope approach) based on climatic forcing that drives the JSBACH model simulation instead of CRU CL2.0 and then make comparisons?

11. Figure 2 caption, the tag (a) and (b) should be switched

12. P19, 1st paragraph, plant uncontrollable N loss pathways, such as DON and fire losses worth mentioning. You can find the discussion about how plant uncontrollable vs. controllable N losses regulate terrestrial N limitation from the modeling perspective in Thomas et al., [2015]. You can also find an example of a global C-N model with DON loss from Gerber et al., [2010]. ãĂĂ 13. P19, 2nd paragraph. I agree BNF is critical in the general terrestrial N cycle simulation, but remains largely unresolved. Gerber et al., [2010] has a more dynamic BNF scheme which takes into N supply, N demand and light availability compared to the NPP or ET approach, but more studies are needed to improve BNF.

Literature cited: Gerber, S., L. O. Hedin, M. Oppenheimer, S. W. Pacala, and E. Shevliakova (2010), Nitrogen cycling and feedbacks in a global dynamic land model, Global Biogeochemical Cycles, 24, doi:10.1029/2008gb003336 Thomas, R. Q., E. N. J. Brookshire, and S. Gerber (2015), Nitrogen limitation on land: how can it occur in Earth system models?, Global Change Biology, 21(5), 1777-1793, doi:10.1111/gcb.12813

---

## Referee Comment (RC2) · Anonymous Referee #2 · 7 Mar 2017

This manuscript is a description of global nitrogen cycling and the reporting of standard climate and CO2 feedback parameters from an Earth system model in preparation for CMIP6 simulations. The advancement in modeling is the addition of N cycling to a previously C only soil decomposition model (YASSO). The unique attribute of the overall modeling framework, with respect to the N cycle, is the assumption that N limitation of plant and microbial activity is not present during the pre-industrial simulation (the CNL assumption). Therefore, the changes in N limitation using the 20 and 21 century simulations are caused by rising atmospheric CO2 concentrations.

It is difficult to tell whether the disagreement between the N gas loss fraction simulation and estimates is due to differences in the climate driver data or due to the ESM. An

offline simulation is needed that uses historical climate driver rather than the coupled model climate drivers to better evaluate the model. Considering the N15-based gas loss fraction is the only spatial evaluation of the model in this manuscript, the simulations used to compare to the observations should allow the most 'clean' comparison possible. Furthermore, more discussion of the uncertainty in the 15N based estimates of N gas loss fraction is needed. How good of a tool is it for evaluating the model?

More discussion is needed about how the CNL assumption influences the results. It appears that the CNL assumption is achieved by reducing the N limitation during the pre-industrial spin-up but it is unclear how this spin-up process influences the overall predictions.

More detailed comments:

Page 1, Line 6: little r is used here but big R is used later on Page 12, Line 19. Which is correct?

Page 1, Line 12: How does CO2 enhance the turnover of organic nitrogen? This result does not seem to be highlighted explicitly elsewhere. In fact, Page 18, Line 19 states the JSBACH is unable to account for the simulation of organic matter turnover through priming.

Page 2, Line 12: 'The exchange of the former. . ." is awkward to read (did YASSO do the exchanging?).

Page 2, Line 18: Add an 'of' between 'recycling' and 'nitrogen'

Page 2, Line 20: Recommend citing Thomas et al. 2015 (Global Change Biology) here

Page 2, Line 21: The Luo et al 2004 is about progressive nitrogen limitation rather than just nitrogen limitation. I recommend adjusting the language

Page 2 Line 23: I recommend adding Zhu and Riley 2015 (Nature Climate Change) to complement the Houlton citation.

[Figure]

Page 3 Line 12: The manuscript the terms 'litter size classes', 'size classes', and 'litter class') are both used. Please be consistent. (see Page 4 line 17 and Table 1 as examples)

Page 4, Equation 1. The matrix is missing the humus pool despite being referenced in the prior sentence

Table 1. Including the decomposition rates would be useful for understanding how the rate constants compare to other models. Is this a fast turover model?

Section 2.1.1: The exact approach to plant microbe competition needs to described in this section. Do microbes have first access? Overall, the manuscript needs a better description of the order of operations for the nitrogen cycle.

Section 2.1.2. I found the w and g subscript to be confusing. What do the w and g stand for?

Section 2.1.2: It seems that the C:N ratio of non- lignified litter is constant across the globe. Does this mean that the C:N ratio of non-lignified biomass is constant across the globe or is there variable retranslocation? An assumption that the C:N non-lignified litter is constant seems to be ignoring known differences in foliar N across forest types.

Page 9, Line 2: Please expand on what criteria was used to tune the parameter. What does it mean that the 'assumption of regarding the absence of CNL in the pre-industrial state is met'?

Section 2.5.2. It should be explicitly stated in this section that the authors reanalyzed existing N15 data. Also, how is the data publically available? Is there a database? Overall more description is needed of the dataset that was used. How is the dataset and analysis similar and different from the Houlton 2015 analysis?

Page 12, Line 3. The sentence states that the mineral nitrogen stocks were in the wide range of estimates but Table 5 does not provide any global estimates for the Mineral nitrogen pools.

Page 12, Line 17. Please provide more information on the consistency between the results used in the manuscript and the Houlton results.

Figure 1. I recommend including a 1:1 plot (simulated vs. reconstructed) as well. It will help the reader understand the bias of the model better.

Figure 4. I like this figure and find it helpful for visualizing the changes to the N cycle.

Page 16, line 32. Please expand on the statement that the overall behavior is in line with mechanisms in Niu et al. 2016. The connection between the model in the manuscript and the conceptual model isn't clear. What does it mean to be 'in line with'?

Figure 5, Why are the units on Figure 5c (kgC) different from the rest of the units (gC) on the figure?

Page 18, line 12. How does the finding illustrate the need of a multitude of carbon-nitrogen models? Please expand on this statement.

Page 18, Lines 28: How does elevated $CO_2$ directly increase respiration? Are you referring the increasing respiration is a requirement to prevent labile C from building up in the plant? Overall, this statement is confusing and needs to be expanded on.

---

## Author Comment (AC1) · 5 Apr 2017

The paper of Goll et al. focuses on the carbon-nitrogen interactions in the new version of JSBACH with updated soil organic matter decomposition model and N component. As nutrient limitation plays an important role in land carbon cycles, the work of Goll et al. is very important towards better predicting of future carbon dynamics. The work is novel in making use of nitrogen isotope data to evaluate process based N simulations. The study is generally well conducted and sufficient for recording model behavior, but I have several concerns or questions listed below.

1. **To what extent the C-N interactions produced from this paper are reliable?**

There is a large uncertainty in N cycle, which makes the C-N interactions difficult to constrain. The N limitation on carbon cycle, or C-N interaction strength, is based on assumptions of CO2 induced nutrient limitation (CNL) from this paper. The assumption of marginal nitrogen effect on pre-industrial C cycle is debatable.

> **Reply:** The concept of CNL is a carbon centered approach, thereby limiting the applicability of the model. We believe it is an appropriate method to reduce the risk of overestimating the land carbon uptake under increasing CO2 concentrations by adding an additional constraint with a minimum increase in the model complexity. We added a discussion of the limitation of the CNL approach to the limitation section (see below)

Although the model can be parameterized based on preindustrial conditions (which may have large uncertainties) that have already taking into account of the N effect, it may misrepresent important mechanisms that regulate long term N effects on C. For example, losses of plant uncontrollable nitrogen, such as through fire, erosion, dissolved organic matter, constrains long term N availability and therefore N impacts on C [Thomas et al.,

2015]. Plant uncontrollable nitrogen loss pathways are not represented in the current version of JSBACH.

> **Reply:** The reviewer makes an important point about the importance of uncontrollable losses. We want to stress that JSBACH does account for uncontrollable losses due to leaching, denitrification (see equations 15&16), as well as fire and LULCC. Changing environmental conditions can cause changes in these loss terms. However, due to the our concept, these losses do not significantly affect the global terrestrial carbon balance under pre-industrial conditions in JSBACH as the referee correctly states. In this point we agree with the referee.
>
> In this study losses by fire and LULCC are omitted due to the simulation setup (1%CO2 simulation) as stated on Page 11 Lines 17-22. Erosion losses of soil organic matter have yet to be incorporated into ESM as stated on Page 19 line 10.

Therefore nitrogen limitation cannot be maintained with strong biological controls on N losses and inputs in JSBACH in the long run, but that does not mean there is no N limitation in the long run in a real world. It may capture transient CO2 responses.

> **Reply**: We agree with the reviewer that the concept of CNL can only be applied to scenarios in which increasing CO2 is the major agent of change. The model is able to capture positive effects of increasing N availability (N deposition , warming stimulated mineralisation) only if the increase in CO2 has increased N demand above the the background availability in the first place. Positive effects like found in Esser et al (2007) or Warlind et al (2014) can not be captured by the model. However, in general models tend to simulate negative effects of N which outweigh the positive ones in projection for the 21st century.
>
> We revised the manuscript to clarify the mentioned shortcomings of the CNL approach in JSBACH.
> We added:
> *P21L31-P22L6: added "Due to the concept of CO2-induced nutrient limitation in JSBACH the nitrogen cycle serves primarily as an additional constraint on the carbon uptake. The advantage of the approach is its low complexity and avoidance of assumptions about the initial state of nutrient limitation thereby taking into account (1) the lack of data regarding the nitrogen cycle (Zaehle, 2013) as well as (2) the large uncertainty about the nutrient constraint on plant productivity (Elser et al., 2007, Zaehle, 2013). The shortcomings of this approach are that it limits the applicability of the model to carbon cycle projections for scenarios of increasing atmospheric CO2 and that it cannot capture any stimulation of the plant productivity due to changes in nitrogen availability itself: In addition to direct increase in nitrogen availability by nitrogen deposition and fertilization, a stimulation of plant productivity can occur due to reduced losses of nitrogen by pathways which are not under control of biota, like fire, leaching, or erosion(Thomas et al., 2015). As a result, the model might underestimate the importance of nitrogen cycling for carbon uptake under elevated CO2."*

As mentioned by the authors, it may misrepresent climate response and potentially the C-N interactions and other aspects that affect C-N interactions. I suggest the authors be cautious about reaching a conclusion about getting the decomposition of soil carbon right first before incorporating C-N interactions as the evaluation should be based on the right representation of C-N interactions and compared to the "true" observation.

> **Reply:** We agree with the referee that this statement cannot be fully backed by the finding of single model, which might not capture the "truth" in particular as it is prone to underestimate the effect of nitrogen on the carbon cycle. We changed the conclusion in abstract to
>
> P1L15-18"These changes are primarily due to the new decomposition model, indicating the importance of soil organic matter decomposition for land carbon feedbacks."

2. How does reproducing the relative fraction of nitrogen loss pathways affect land carbon? **Or how does C-N models benefit from an accurate representation of the relative N loss pathways?**

It seems to me the focus of this paper is on C-N interaction. Does accurate representation of the relative N loss pathway (leaching vs. gaseous) help in correcting C-N interactions? It is possible to have a correct leaching: gaseous loss ratio while have a wrong simulation of leaching loss or gaseous loss. As the ratio can be tuned through parameters, such as the fraction of soil water lost to rivers per day, the fraction of mineral nitrogen in soil solution, and fdenit estimation from 15N relies strongly on climatic conditions, a reasonable representation of the spatial pattern of fdenit does not necessarily mean a good simulation of mineral N and N limitation on biological activities. It makes the 15N based evaluation more valuable if the authors can clarify the merits of such evaluation for the general N cycling and C-N interaction.

> **Reply:** δ15N measurements are one of the few sources of spatially extensive data relevant to the N cycle (Houlton et al., 2015). But as the referee states correctly this information does not tell about the total loss rate (or throughput) of N, which is almost impossible to measure (because a large part of the denitrification loss is in the form of N2). Despite this shortcoming of the data, the good spatial agreement between observed and model loss pathway shows that the model captures the difference in the respective environmental controls on the losses processes. As these losses are not directly under control of vegetation (see point above), it is important to ensure that their sensitivity to climatic changes is realistically implemented in the model (here by substituting space with time). It is not given that ESMs can capture the loss pathways see for example Houghton et al.,2015., in particular not with such high success.

We added to the manuscript a discussion of the benefits and shortcomings of the δ15N measurements product, and stress that the fraction of water lost is not tuned but calculated by a soil hydrological scheme:

P12L25: added " δ15N  data measurements are one of the few sources of spatially extensive data relevant to the nitrogen cycle, "

P14L30/31: added "However, this comparison does not allow to draw any conclusion about the magnitude of the losses."

P14L32-P15L2: added: " The  reconstructed f_denit from Houlton et al. (2015)maps
 (Figure (A1&2) presented here are generally similar to those presented
 by Houlton et al. (2015), with high fractions (ca 80%) in the tropics and mid-latitude deserts, a strong gradient of decreasing  fractions with decreasing temperature towards high altitudes and latitudes, and values in the range 0-20% reached in
 cold, wet climates in the north. For a detailed discussion of differences see SI."

P10L10-11: added" *$f_{h2o}$ is computed dynamically accounting for evapotranspiration, precipitation, and changes in the soil water storage using a 5 layer soil hydrological scheme (Hagemann et al. 2014).*"

The referee is right that the good agreement between observed and simulated loss pathways does not allow any conclusion about N limitation of vegetation, but we cannot find such a statement in the manuscript. No changes done.

3. **Model description is not very clear and is confusing in some parts.**
 As this paper focus on how N affects land carbon simulation, it is better to let readers know how N limitation regulates photosynthesis (GPP or NPP) and organic matter decomposition which are two key nexus points in C-N interactions. As the model descriptions combine terms from the YASSO model, the old JSBACH model and the updated JSBACH model, it is difficult to follow especially when the structures of these models are not the same. I suggest reworking on model description. More detailed suggestions are available in Minor Points.

**Reply:** The manuscript reports the changes in the soil part of the nitrogen as the rest is described in detail in Goll et al. (2012) and Parida (2010).  We agree that an overview of the C and N cycling and its interactions is beneficial for the reader. We therefore added a scheme (Figure 1) showing the interactions between the N and the C cycle with a comprehensive caption explaining the C-N interactions:

Added P5:
*"Figure 1. Schematic representation of carbon (top) and nitrogen (below) cycling in JSBACH. Vegetation is represented by four pools: "active" (leaves and non-lignified tissue) and "wood" (stem and branches), "reserve" (sugar and starches) and "mobile"*

*(labile nitrogen) (Goll et al., 2012). Dead organic matter is represented by "non-lignified litter", "lignified litter" (lignified litter and fast-decomposing soil organic matter), and "humus" (slow-decomposing organic matter) (Raddatz et al., 2007). All organic matter pools have fixed C:N ratios, except the pools "reserve", "labile" and "non-lignified litter". While the first two pools have no corresponding pool, the C:N ratio of the latter pool varies according the balance between immobilization demand and supply. The carbon in the litter compartment is further refined into the acid-soluble (A), water-soluble (W), ethanol-soluble (E), and non-soluble (N) compounds (Goll et al., 2015) which have no C:N ratio assigned. Soil mineral nitrogen is represented by a single pool (soil mineral pool). The opposing triangle marks carbon fluxes which are downregulated in case the nitrogen demand exceeds the nitrogen supply."*

*P3L9-10: added: " a scheme of the cycling of carbon and nitrogen as well as their interactions are given in Figure 1.*

We further included in the appendix the calculation of the nitrogen limitation factor:
P24/25: added:
*"**The nitrogen limitation factor***
*The nitrogen limitation factor, $f^{N}_{limit}$, is calculated based on a supply and demand approach (Parida 2011, Goll et al., 2012). In a first step, potential carbon fluxes are computed from which the gross mineralisation, immobilization ($D_{micr}$) and plant uptake of mineral nitrogen ($D_{veg}$) is diagnosed.*
*In a second step, all fluxes consuming nitrogen (donor compartment has a higher C:N ratio than the receiving pool as well as plant uptake) are down-regulated in case nitrogen demand cannot be met by the nitrogen supply. Hereby, a common scalar ($f^{N}_{limit}$) is used thereby no assumption about the relative competitive strengths of microbial and plant consumption has to be made.*
*<<< EQUATION: SEE PDF >>>*
*where the term in square bracket is the maximum rate at which the soil mineral nitrogen pool can supply nitrogen. Note that in the discretized formulation the mineral nitrogen pool can at most be depleted during a single model time step ($\Delta t$). We thus set this maximum rate to $\frac{dN_{smin}}{\Delta t}$. "*

Please see also the changes done in response to referee #2 points about the model description section.

Minor Points:

1. Is it appropriate to have many citations in abstract?
   **Reply:** we removed all citations from the abstract

2. Page 1, line 10, the reference of Shi et al., 2015 is not relevant. Do you mean Shi Z, Yang Y, Luo Y, Zhou X, Weng E, Finzi A. 2015. Inverse analysis of coupled carbonnitrogen cycles against multiple datasets at ambient and elevated CO2. Journal of Plant Ecology, doi:10.1093/jpe/rtv059

**Reply:** Yes, we meant Shi et al. 2016; we corrected the references at P16L19 and removed the one mentioned here.

3. P4,L5, equation 1, line 9, is there a H component in the matrix equation?
**Reply**: There is no H component in the matrix equation. We added the equation EQ3 for the dynamics of the Humus pool with its description and corrected the text.

*P4L11-12: revised "Matrix $C$ describes the soil carbon pools (A, W, E, N ) of the two litter size classes ($i$) in JSBACH, excluding recalcitrant humic substances ($C_H$):"*

*P4L17-P5L1: added: "The dynamics of the humus pool ($C_H$) are described as:*
*<<<EQUATION: SEE PDF >><*
*where $p_H$ is the relative mass flow parameter and $k_H$ the decomposition rate of the humus pool. "*

4. P5, lines 2-3 "lignified litter and fast decomposing organic matter" is confusing as no "fast decomposing organic matter" is mentioned in the carbon part.

**Reply:** We revised the text to avoid confusions.
P6L11-P7L5: revised/added: "Nitrogen in litter and soil organic matter is separated into three pools, namely slowly-decomposing organic matter $C_s$, lignified litter and fast decomposing organic matter $C_{lw}$, as well as non-lignified litter and fast decomposing organic matter $C_{la}$ (Goll et al.,2012). We assigned each of the three nitrogen pools to one or more corresponding YASSO pools (Table~\ref{tab:corr}). A refinement of the representation of nitrogen in decomposing material following strictly the carbon classification is not straightforward as the carbon pools ($A,W,E,N$) defined by their respective solubility characteristics do not correspond to substance classes with distinguished stoichiometries."

5. P7, equation 10, where is the nitrogen flux from your la class (non-lignified &fast decomposing organic matter)? The third term, the lignified flux is not clear. Why do you have (rw-rlw)*F for lignified flux while only have rla*F for non-lignified flux in equation 7? The description from Lines 20-21 is not clear. Why do you differentiate N-to-C ratio of lignified litter and biomass?

**Reply:** We added two sentences to clarify the two mentioned points
Added:
P9L6-11: added: *"Due to the lower nitrogen content of litter compared to humus, the decomposition of lignified and non-lignified litter corresponds to a net immobilization of nitrogen, which is part of the $D_{micr}$. The term $(r_w - r_{lw})F^C_{w->lw}$ represents nitrogen leaching from freshly shedded wood given by the decomposition and the stoichiometries assigned to wood ($r_w$ ) and lignified litter ($r_{lw}$)."*

6. P9, L10, "cchange" to "change "

**Reply:** typo corrected

7. Section 2.5.2 Nitrogen loss pathway data. I may have missed some part, but the description on how to estimate fdenit is not very clear. You fitted data to equation 19. So in equation 19, what are known and what are need to be estimated? If k is the only factor need to be estimated, what is the purpose to estimate k as equation 20 based on which to estimate fdenit does not need k

> **reply**: In the equation 19, we need to estimate the *k* **AND** *ε* (gaseous discrimination factor) from soil δ15N using non-linear least-squares regression method. By re-arrenging Eq19 and 20 we can derive f_denitr. We revised the text to clarify this.
>
> Changes in the text:
>
> P13L20: revised "*The data were then fitted via epsilon the gaseous discrimination factor and a constant k by non-linear least-squares regression to the relationship*"
>
> P13L26: added: ". *Re-arranging Equation 19 and 20 we get f_denitr=(1+k(f(q)/f(T)))-1*"
>
> See also reply to Referee#2 and revisions on line P12L25-P13L2 which gives additional information regarding the methodology and major differences to approach applied by Houlton et al. (2015).

8. P12, L3-5 No compiled mineral N in Table 5 is available for comparison and indicates simulated mineral nitrogen stock is within the range of estimates. There is no available data in Table 5 to compare denitrification between simulated 1850 vs. observation-based 1850. Comparing between simulated value at 1850 with present is not appropriate as nitrogen cycle is altered strongly by anthropogenic activities since the industrialization.

> **Reply**: we agree with the reviewer about the human impact on the nitrogen cycle. The estimates compiled in Table5 are all estimates we are aware of. We removed the the statement about denitrification and stress that estimates are not available for all variables.
>
> We revised the text:
> P14L6-10: revised: "*The model simulates nitrogen stocks and fluxes under pre-industrial conditions well within the wide range of the few available observation based estimates (Table 5). Most of the estimates are for present day conditions and thus are not directly comparable due to the human influence on the nitrogen cycle (Galloway et al., 2013).* "

9. P12,L6, Is nitrogen in la (non-lignified litter & fast decomposing organic matter) part of the organic nitrogen stocks? Equation 7 says it is not prescribed based on C:N.

**Reply:** We added a table with the simulated CN ratios of non-lignified litter & fast decomposing organic matter in comparison with observation to the appendix and state in the main text :

P14L12-13: Added: *"except for non-lignified litter and fast decomposing soil organic matter which shows in general good agreement with observed C:N ratios (see appendix)."*

Added "appendix B" including a Table A1 to appendix :

P26: added: "**Evaluation of dynamically computed C:N ratios**
*The only ecosystem compartment in JSBACH which has a flexible stoichiometry is non-lignified litter and fast-decomposing organic matter.*
*The simulated carbon to nitrogen ratios of this compartment for the six plant functional types in JSBACH are in rather good agreement with observations of foliage litter from the ART-DECO database (Table~\ref{tab:CN}),*
*except for tropical broadleaved deciduous trees and extra-tropical evergreen trees.*
*Reasons for the overestimation of nitrogen content in litter from extratropical evergreen tree is the global parametrization of leaf stoichiometry applied in JSBACH which does not capture the lower leaf nitrogen concentration in needle-leaved trees compared to broad-leaved trees (Kattge et al., 2011).*
*The data for tropical species is very scarce and the variability among species is large, which hamper the interpretation of the mismatch between model and observation for the tropical broadleaved deciduous trees."*

10. P13, if climatic forcing is the reason for mismatch, is it feasible to calculate fdenit (the isotope approach) based on climatic forcing that drives the JSBACH model simulation instead of CRU CL2.0 and then make comparisons?

**Reply:** According to the referee's suggestion, we recalculated f_denit map using the climatology from the model in addition to observed climatology. We now use the f_denit from simulated climate in the main manuscript and move the map of f_denit derived from observed climatology to the appendix. We modified the method section, the discussion section, the conclusion section, and the appendix. We added Figure A1 showing the f_denit derived from observed climatology and a comparison of the frequency distributions for the two approaches.

P13L28-P14L3: revised: *"A spatial map of $f_{denit}$ was derived from the empirical relationship between temperature, runoff and $f_{denit}$ using simulated values of $f(q)$ and $f_m(T_m)$ from JSBACH. Thereby, model biases in climate are accounted for in the data derived $f_{denit}$ which allows a straightforward comparison with simulated $f_{denit}$. In addition, we derived maps of $f_{denit}$ based on monthly grids of observed mean climate from 1961--1990 covering the global land surface at a 10 \unit{minute} spatial resolution (CRU CL2.0) \citep{New2002} which are shown in the appendix."*

P15L3-P15L18: revised: *"In comparison with the reconstructed fractional gaseous loss from simulated climate (Figure 2a), we find that the model is in good agreement*

*(Pearson R=0.76, RMSE=0.2, Taylor score=0.83). The model underestimates high values of $f_{denit}$ and overestimate low values  Figure A2). In regions with cold winter temperatures where  enitrification losses are small the model overestimates denitrification losses  Figure 2c) These model biases likely derive from the simplistic representation of denitrification  as a function of soil moisture and substrate availability, which omits effects of temperature (Butterbach-Bahl et al., 2013). Additionally, other omitted factors like oxygen concentration, soil pH, mineralogy, and transport processes (Butterbach-Bahl et al., 2013) might contribute to the bias.*
"

P23L6-16: revised:  *"The simulated response of primary productivity to increasing CO2,  simulated litter stoichiometries, as well as the simulated spatial variability in nitrogen loss pathways  are in good agreement with observation based estimates. Here we show that a simple representation of mineral nitrogen dynamics can achieve a high agreement with observation in respect to nitrogen loss pathways.*
*Further refinements of denitrification should address the relationship between denitrification and low soil moisture availability and as well as introduce a temperature scaling function."*

P26 added:
"**Consistency of nitrogen loss pathways with earlier estimates**
*The reconstructed $f_{denit}$ map from observed climatology (Figure A1)  is generally similar to one presented by Houlton et al. (2015), with high fractions (ca 80%) in the tropics and mid-latitude deserts, a strong gradient of decreasing fractions with decreasing temperature towards high altitudes and latitudes,  and values in the range 0-20% reached in cold, wet climates in the north.  However, some differences are apparent, most obviously connected with the use of mean annual temperature (MAT) by Houlton et al. (2015) to index microbial activity.  MAT becomes extremely low in Eurasia towards the northeast, for example, and accordingly, Houlton et al. (2015)estimates of the denitrification fraction become very low there.*
*Craine et al. (2015) noted that climates with very low MAT (including sites in NE Siberia) showed anomalous values of soil $\delta^{15}N$, more similar to those of warmer climates. Our approach takes account of  this by the use of an index that is much more responsive to the warm summers than to the extreme cold winters found in hypercontinental climates.*

*When simulated climatology is used to upscale the empirical relationship between temperature, runoff and soil $\delta^{15}N$,  the influence of biases in simulated climatology on $f_{denit}$ become apparent. The overestimation of precipitation and subsequently runoff of about 20%  in MPI-ESM (Weedon et al., 2011, Hagemann & Stacke, 2013} leads to a pronounced peak in the histogram of $f_{denit}$ at about 0.1-0.2 (Figure A1),  which is mostly in the mid and high latitudes regions in northern hemisphere."*

11. Figure 2 caption, the tag (a) and (b) should be switched

**Reply:** corrected

12. P19, 1st paragraph, plant uncontrollable N loss pathways, such as DON and fire losses worth mentioning. You can find the discussion about how plant uncontrollable vs. controllable N losses regulate terrestrial N limitation from the modeling perspective in Thomas et al., [2015]. You can also find an example of a global C-N model with DON loss from Gerber et al., [2010].

  **Reply:**  see text revision as a response to the major point of critic #1 (above)

13. P19, 2nd paragraph. I agree BNF is critical  in the general terrestrial N cycle simulation, but remains largely unresolved. Gerber et al., [2010] has a more dynamic BNF scheme which takes into N supply, N demand and light availability compared to the NPP or ET approach, but more studies are needed to improve BNF.

  **Reply:** we added references to to more sophisticated BNF models and rephrased the sentence to avoid implying there is no better model around than used in JSBACH:

  *P22L16-18: revised "BNF models which better resolve the governing mechanisms,*
  *for example (Gerber e al., 2010, Fisher et al., 2012), should be incorporated into*
  *ESMs to increase the reliability of the simulated pace of changes in BNF (Meyerholt*
  *& Zaehle, 2016).*
This manuscript is a description of global nitrogen cycling and the reporting of standard climate and CO2 feedback parameters from an Earth system model in preparation for CMIP6 simulations. The advancement in modeling is the addition of N cycling to a previously

C only soil decomposition model (YASSO). The unique attribute of the overall modeling framework, with respect to the N cycle, is the assumption that N limitation of plant and microbial activity is not present during the pre-industrial simulation (the CNL assumption). Therefore, the changes in N limitation using the 20 and 21 century simulations are caused by rising atmospheric $CO_2$ concentrations.

It is difficult to tell whether the disagreement between the N gas loss fraction simulation and estimates is due to differences in the climate driver data or due to the ESM. An offline simulation is needed that uses historical climate driver rather than the coupled model climate drivers to better evaluate the model. Considering the N15-based gas loss fraction is the only spatial evaluation of the model in this manuscript, the simulations used to compare to the observations should allow the most 'clean' comparison possible.

> **Reply:** We agree with the Referee about to need to "clean" the comparison. Unfortunately, it is computational not feasible to repeat the JSBACH simulations, however a clean comparison can also be achieved by recalculating the observed fdenit using MPI-ESM climate as Referee #1 suggested. The latter is feasible. In the following we repeat the answer to Referee #1 and the according changes to the manuscript:

<< COPY

> According to the referee's suggestion, we recalculated f_denit map using the climatology from the model in addition to observed climatology. We now use the f_denit from simulated climate in the main manuscript and move the map of f_denit derived from observed climatology to the appendix. We modified the method section, the discussion section, the conclusion section, and the appendix.

> P13L7-10: revised: *"A spatial map of $f_{denit}$ was derived from the empirical relationship between temperature, runoff and $f_{denit}$ using simulated values of $f(q)$ and $f_m(T_m)$ from JSBACH. Thereby, model biases in climate are accounted for in the data derived $f_{denit}$ which allows a straightforward comparison with simulated $f_{denit}$. In addition, we derived maps of $f_{denit}$ based on monthly grids of observed mean climate from 1961--1990 covering the global land surface at a 10 \unit{minute} spatial resolution (CRU CL2.0) \citep{New2002} which are shown in the appendix."*

> P15L3-18: revised: *"In comparison with the reconstructed fractional gaseous loss from simulated climate (Figure 2a), we find that the model is in good agreement (Pearson R=0.76, RMSE=0.2, Taylor score=0.83). The model underestimates high values of $f_{denit}$ and overestimate low values Figure A2). In regions with cold winter temperatures where enitrification losses are small the model overestimates denitrification losses Figure 2c) These model biases likely derive from the simplistic representation of denitrification as a function of soil moisture and substrate availability, which omits effects of temperature (Butterbach-Bahl et al., 2013). Additionally, other omitted factors like oxygen concentration, soil pH, mineralogy, and transport processes (Butterbach-Bahl et al., 2013) might contribute to the bias.*

"

P23L6-16: revised: *"The simulated response of primary productivity to increasing CO2, simulated litter stoichiometries, as well as the simulated spatial variability in nitrogen loss pathways are in good agreement with observation based estimates. Here we show that a simple representation of mineral nitrogen dynamics can achieve a high agreement with observation in respect to nitrogen loss pathways. Further refinements of denitrification should address the relationship between denitrification and low soil moisture availability and as well as introduce a temperature scaling function."*

P26: added:
"**Consistency of nitrogen loss pathways with earlier estimates**
*The reconstructed $f_{denit}$ map from observed climatology (Figure A1) is generally similar to one presented by Houlton et al. (2015), with high fractions (ca 80%) in the tropics and mid-latitude deserts, a strong gradient of decreasing fractions with decreasing temperature towards high altitudes and latitudes, and values in the range 0-20% reached in cold, wet climates in the north. However, some differences are apparent, most obviously connected with the use of mean annual temperature (MAT) by Houlton et al. (2015) to index microbial activity. MAT becomes extremely low in Eurasia towards the northeast, for example, and accordingly, Houlton et al. (2015)estimates of the denitrification fraction become very low there.*
*Craine et al. (2015) noted that climates with very low MAT (including sites in NE Siberia) showed anomalous values of soil $\delta^{15}$N, more similar to those of warmer climates. Our approach takes account of this by the use of an index that is much more responsive to the warm summers than to the extreme cold winters found in hypercontinental climates.*

*When simulated climatology is used to upscale the empirical relationship between temperature, runoff and soil $\delta^{15}$N, the influence of biases in simulated climatology on $f_{denit}$ become apparent. The overestimation of precipitation and subsequently runoff of about 20% in MPI-ESM (Weedon et al., 2011, Hagemann & Stacke, 2013} leads to a pronounced peak in the histogram of $f_{denit}$ at about 0.1-0.2 (Figure A1), which is mostly in the mid and high latitudes regions in northern hemisphere."*

COPY >>

Furthermore, more discussion of the uncertainty in the 15N based estimates of N gas loss fraction is needed. How good of a tool is it for evaluating the model?

> **Reply:** We added a discussion about the advantages and shortcomings of using 15N based estimates of N gas loss to the manuscript, as well as discuss the differences to an earlier estimate of Houlton et al. (2015)
>
> P14L24/25: added ", which are one of the few sources of spatially extensive data relevant to the nitrogen cycle, "

P14L30/31 added "However, this comparison does not allow to draw any conclusion about the magnitude of the losses."

P14L32-P15L2: added: " The reconstructed f_denit from Houlton et al. (2015)maps
 (Figure (A1&2) presented here are generally similar to those presented by Houlton et al. (2015), with high fractions (ca 80%) in the tropics and mid-latitude deserts, a strong gradient of decreasing fractions with decreasing temperature towards high altitudes and latitudes, and values in the range 0-20% reached in cold, wet climates in the north. For a detailed discussion of differences see SI."

More discussion is needed about how the CNL assumption influences the results. It appears that the CNL assumption is achieved by reducing the N limitation during the pre-industrial spin-up but it is unclear how this spin-up process influences the overall predictions.

**Reply:** We added a discussion of the implications of the CNL approach on the results:

*P21L31-P22L7: added "Due to the concept of CO2-induced nutrient limitation in JSBACH the nitrogen cycle serves primarily as an additional constraint on the carbon uptake. The advantage of the approach is its low complexity and avoidance of assumptions about the initial state of nutrient limitation thereby taking into account (1) the lack of data regarding the nitrogen cycle (Zaehle, 2013) as well as (2) the large uncertainty about the nutrient constraint on plant productivity (Elser et al., 2007, Zaehle, 2013). The shortcomings of this approach are that it limits the applicability of the model to carbon cycle projections for scenarios of increasing atmospheric CO2 and that it cannot capture any stimulation of the plant productivity due to changes in nitrogen availability itself: In addition to direct increase in nitrogen availability by nitrogen deposition and fertilization, a stimulation of plant productivity can occur due to reduced losses of nitrogen by pathways which are not under control of biota, like fire, leaching, or erosion(Thomas et al., 2015). As a result, the model might underestimate the importance of nitrogen cycling for carbon uptake under elevated CO2."*

**More detailed comments:**

Page 1, Line 6: little r is used here but big R is used later on Page 12, Line 19. Which is correct?
**Reply:** corrected P1L6

Page 1, Line 12: How does CO2 enhance the turnover of organic nitrogen? This result does not seem to be highlighted explicitly elsewhere. In fact, Page 18, Line 19 states the JSBACH is unable to account for the simulation of organic matter turnover through priming.
**Reply:** corrected. Turnover is enhanced due to warming not CO2.

P1L12: revised :*"In line with evidence from elevated carbon dioxide manipulation experiments, pronounced nitrogen scarcity is alleviated by the accumulation of nitrogen due to enhanced nitrogen inputs by biological nitrogen fixation and reduced losses by leaching and volatilization. The stimulation of turnover of organic nitrogen by increasing temperatures further counteracts scarcity."*

Page 2, Line 12: 'The exchange of the former. . .' is awkward to read (did YASSO do the exchanging?).
**Reply:** corrected. P2L13: change "by the YASSO model" to "with the YASSO model"

Page 2, Line 18: Add an 'of' between 'recycling' and 'nitrogen'
**Reply:** corrected P2L18

Page 2, Line 20: Recommend citing Thomas et al. 2015 (Global Change Biology) here
**Reply:** reference added P2L20

Page 2, Line 21: The Luo et al 2004 is about progressive nitrogen limitation rather than just nitrogen limitation. I recommend adjusting the language
**Reply:** added "(progressive)" P2L22

Page 2 Line 23: I recommend adding Zhu and Riley 2015 (Nature Climate Change) to complement the Houlton citation.
**Reply:** reference added. P2L23

Page 3 Line 12: The manuscript the terms 'litter size classes', 'size classes', and 'litter class') are both used. Please be consistent. (see Page 4 line 17 and Table 1 as examples)
**Reply:** We revised throughout the text "size class" and "litter size class" to "litter class" and added the following information:
P3L11-15: added /revised: "*The decomposition model (YASSO) is based on a compilation of litter decomposition and soil carbon data and distinguish organic matter fractions according to litter size and solubility (Tuomi et al., 2008,2009, 2011). In JSBACH we use two litter size classes, which correspond to litter from non-lignified and lignified plant material (Goll et al., 2015). Each of the two litter classes is further refined into four solubility classes*"

Page 4, Equation 1. The matrix is missing the humus pool despite being referenced in the prior sentence
**Reply**: There is no H component in the matrix equation. We added the equation EQ3 for the dynamics of the Humus pool with its description and corrected the text.

*P4L11-12: revised "Matrix $C$ describes the soil carbon pools (A, W, E, N ) of the two litter size classes ($i$) in JSBACH, excluding recalcitrant humic substances ($C_H$):"*

P4L17-P4L20: added: *"The dynamics of the humus pool ($C\_H$) are described as: <<< EQUATION: SEE PDF >>>*
*where $p\_H$ is the relative mass flow parameter and $k\_H$ the decomposition rate of the humus pool. "*

Table 1. Including the decomposition rates would be useful for understanding how the rate constants compare to other models. Is this a fast turover model?

**Reply:** We added mineralisation rates and biomass nitrogen and carbon to Table 5 and discuss the added variables in text.

P14L14-18: added *"Mineralisation of organic nitrogen is the major source of nitrogen for vegetation and the simulated flux is less than existing model based estimates for present day ranging between 980--1030 (Smith et al. 2014, Zaehle et al. 2010). In models, nitrogen mineralisation is solely a by-product of decomposition of soil organic carbon and we thus attribute the differences between simulated mineralisation to the use of YASSO decomposition model compared to the use of the CENTRUY decomposition model (Smith et al. 2014, Zaehle et al. 2010) as the soil C:N stoichiometries are comparable among models."*

Section 2.1.1: The exact approach to plant microbe competition needs to described in this section. Do microbes have first access?

**Reply:** we revised the methods to add this information:

P6L4-9 revised *"In a first step, potential carbon fluxes are computed from which the release, immobilization and plant uptake of mineral nitrogen is diagnosed. In a second step, all fluxes consuming nitrogen (donor compartment has a higher C:N ratio than the receiving pool) are down-regulated in case nitrogen demand cannot be met by the nitrogen supply. Hereby a common scalar ($f^{N}\_{limit}$) (see appendix) is used thereby no assumption about the relative competitive strengths of microbial and plant consumption has to be made. In case nitrogen demand is met by the supply, the fluxes computed in the first step are taken as actual ones without any modification."*

We further added details on the calculation of the limitation factor to the appendix.

P24/25 added: *"**The nitrogen limitation factor**
The nitrogen limitation factor, $f^{N}\_{limit}$, is calculated based on a supply and demand approach (Parida 2011,Goll et al., 2012). In a first step, potential carbon fluxes are computed from which the gross mineralisation, immobilization ($D\_{micr}$) and plant uptake of mineral nitrogen ($D\_{veg}$) is diagnosed.
In a second step, all fluxes consuming nitrogen (donor compartment has a higher C:N ratio than the receiving pool as well as plant uptake) are down-regulated in case nitrogen demand cannot be met by the nitrogen supply.
Hereby, a common scalar ($f^{N}\_{limit}$) is used thereby no assumption about the relative competitive strengths of microbial and plant consumption has to be made.
<<< EQUATION: SEE PDF >>>*

*where the term in square bracket is the maximum rate at which the soil mineral nitrogen pool can supply nitrogen. Note that in the discretized formulation the mineral nitrogen pool can at most be depleted during a single model time step ($\Delta t$). We thus set this maximum rate to $\frac{dN_{smin}}{\Delta t}$. "*

Overall, the manuscript needs a better description of the order of operations for the nitrogen cycle.

**Reply:**, we now explicitly state that losses are prioritized at each timestep. Plant uptake and immobilisation happens simultaneously. See last reply above

Added P10L5: *"The losses of nitrogen are given priority over immobilization and plant uptake each time step."*

We added the to the discussion:
P22L23-24: added: "pathways is to a large degree in line with δ 15 N-derived patterns, despite the low performance of another sequential competition model (Houlton et al., 2015; Zhu and Riley, 2015)."

Section 2.1.2. I found the w and g subscript to be confusing. What do the w and g stand for?

**Reply:** we exchange subscript 'g' with 'a' for "active" plant tissue. Subscript 'w' stands' for 'woody' plant tissue. We already used the term "active, non-lignified plant tissue" throughout the text. We exchanged the term "lignified plant material" with "lignified (woody) plant material" to make this clear.

Section 2.1.2: It seems that the C:N ratio of non- lignified litter is constant across the globe. Does this mean that the C:N ratio of non-lignified biomass is constant across the globe or is there variable retranslocation? An assumption that the C:N non-lignified litter is constant seems to be ignoring known differences in foliar N across forest types.

**Reply:** we use globally uniform parametrization of stoichiometry and resorption. The resorption of leaf nitrogen is not that flexible (40-60%) (Sterner & Elser 2005, Mc Groddy et al. 2004) compared to other nutrients for example leaf phosphorus (McGroddy et al. 2004). Goll et al. (2012) showed that the effect of stoichiometric parametrization on the effect of nutrients on the carbon uptake is rather small in JSBACH. What matters for the effect of nutrients on the carbon uptake is the plasticity in stoichiometry & resorption not the baseline value itself, see for example Meyerholt et al (2015). Plasticity in plant traits is in general omitted in JSBACH.

We added the implications of the omission of plasticity in nitrogen related plant traits to the limitation section:
P21L27: revised *"The current understanding of processes governing the terrestrial nitrogen balance is still rather limited \citep{Zaehle2013}, and several processes*

*which might be of importance, [...] , plasticity in stoichiometry and leaf nutrient recycling \citep{Zaehle2014,Meyerholt2015}, [...] are not represented in JSBACH."*

Page 9, Line 2: Please expand on what criteria was used to tune the parameter. What does it mean that the 'assumption of regarding the absence of CNL in the pre-industrial state is met'?

**Reply:** clarified P10L24-25: "so that the assumption regarding the absence of CNL in the pre-industrial state is met which equals to a negligible (<2\%)) effect of nitrogen on net primary productivity and carbon stocks"

Section 2.5.2. It should be explicitly stated in this section that the authors reanalyzed existing N15 data. Also, how is the data publically available? Is there a database?

**Reply:** We added a paragraph to the method section to list the differences in the methods. P12/13

**Reply:** we explicitly state in the method section that we reanalyzed published data
*P13L4-7: "predictors fitted to publicly available data on soil δ15N (Patino et al., 2009; Cheng et al., 2009; McCarthy and Pataki, 2010; Fang et al., 2011, 2013; Hilton et al., 2013; Peri et al., 2012; Viani et al., 2011; Sommer et al., 2012; Yi and Yang, 2007)."*

We further make this data available as all other data used in this study as stated in the Data availability section:
*"Primary data and scripts used in the analysis and other supplementary information that may be useful in reproducing the author's work are archived by the Max Planck Institute for Meteorology and can be obtained by contacting* [publications@mpimet.mpg.de](mailto:publications@mpimet.mpg.de)*."*

No changes done.

Overall more description is needed of the dataset that was used. How is the dataset and analysis similar and different from the Houlton 2015 analysis?

**Reply:** We added to methods detailed information on the differences between the two approaches:

P12L25-P13L2: added *"$\delta$15N data measurements are one of the few sources of spatially extensive data relevant to the nitrogen cycle (Houlton et al., 2015) as one can infer information about the nitrogen pathways. Houlton et al. (2015) derived the fraction of nitrogen loss in gaseous form ($f_{denit}$) based on (Amundson et al., 2003) best-fitting multiple regression equation for soil $\delta$15N as a function of mean annual temperature (MAT) and mean annual precipitation (MAP). The data*

*set used to generate this equation consisted of 29 samples, and the coefficient of determination was 0.39. Amundson et al. (2003) remarked that 'pending the availability of more soil $\delta$15N analyses, the present Figure ... represents our best estimate of trends ... in global soil $\delta$15N values' (p. 5). We have updated this analysis in three ways: (a) by including a larger number (659) of soil $\delta$15N samples; (b) by substituting an annually integrated index of temperature-related microbial activity for MAT, and an index of leaching (derived from runoff) for MAP – i.e. using indices more closely related to the governing processes; and (c) by using non-linear regression to fit a statistical model that is explicitly based on the isotopic mass balance equation of (Houlton & Bai, 2009)."*

Page 12, Line 3. The sentence states that the mineral nitrogen stocks were in the wide range of estimates but Table 5 does not provide any global estimates for the Mineral nitrogen pools

**Reply:** no estimates for mineral N are available. We revised the text
P14L6-10: revised: "*The model simulates nitrogen stocks and fluxes under pre-industrial conditions well within the wide range of the few available observation based estimates (Table 5). Most of the estimates are for present day conditions and thus are not directly comparable due to the human influence on the nitrogen cycle (Galloway et al., 2013).* "

Page 12, Line 17. Please provide more information on the consistency between the results used in the manuscript and the Houlton results.

**Reply:** We added a section to the appendix to discuss the differences in respect ot the methodologies:
Added to appendix (Page26): "
 ***Consistency of nitrogen loss pathways with earlier estimates***
*The reconstructed f_{denit} map (Figure~(\ref{SIfig:15N}) presented here is generally similar to one presented by (Houlton et al., 2015), with high fractions (ca 80\%) in the tropics and mid-latitude deserts, a strong gradient of decreasing fractions with decreasing temperature towards high altitudes and latitudes, and values in the range 0-20\% reached in cold, wet climates in the north. However, some differences are apparent, most obviously connected with the use of mean annual temperature (MAT) by Houlton et al. (2015) to index microbial activity. MAT becomes extremely low in Eurasia towards the northeast, for example, and accordingly, Houlton et al. (2015) estimates of the denitrification fraction become very low there. Craine et al. (2015) noted that climates with very low MAT (including sites in NE Siberia) showed anomalous values of soil $\delta^{15}$N, more similar to those of warmer climates. Our approach takes account of this by the use of an index that is much more responsive to the warm summers than to the extreme cold winters found in hypercontinental climates. When simulated climatology is used to upscale the empirical relationship between temperature, runoff and soil $\delta^{15}$N, the influence of biases in simulated climatology on $f_{denit}$ becomes apparent. The*

*overestimation of precipitation and subsequently runoff of about 20\% in MPI-ESM (Weedon et al., 2011, Hagemann & Stacke, 2013) leads to a pronounced peak in the histogram of $f\_denit$ at about 0.1-0.2, which is mostly in the mid-and high latitudes regions in northern hemisphere."*

Figure 1. I recommend including a 1:1 plot (simulated vs. reconstructed) as well. It will help the reader understand the bias of the model better.

    **Reply:** We added 1:1 plot to the appendix Fig A2, and added to the discussion: P15L6-14: added: *"The model underestimates high values of f_denit and overestimate low values (Figure A2)"*

Figure 4. I like this figure and find it helpful for visualizing the changes to the N cycle.

    **Reply:** thanks. No changes needed

 Page 16, line 32. Please expand on the statement that the overall behavior is in line with mechanisms in Niu et al. 2016. The connection between the model in the manuscript and the conceptual model isn't clear. What does it mean to be 'in line with'?

    **Reply:** we explicitly state what we mean with "overall behaviour": P19L30: revised: *"The simulated increase in tightness of the nitrogen cycle as mineral nitrogen stocks deplete is in line with the substrate-based mechanisms "*

Figure 5, Why are the units on Figure 5c (kgC) different from the rest of the units (gC) on the figure?

    **Reply:** We chose to plot kg(C) instead to g(C) in 5c) to avoid small values. No changes done, but if the editor shares the Referee's concern, too, we can change the units.

Page 18, line 12. How does the finding illustrate the need of a multitude of carbon nitrogen models? Please expand on this statement.

    **Reply:** We rephrased the sentence to specify our findings P20L12-P21L2"The contrasting findings regarding the effect of nitrogen on the land carbon feedbacks illustrates the need of a multitude of carbon-nitrogen models to draw general conclusions."

Page 18, Lines 28: How does elevated CO2 directly increase respiration? Are you referring the increasing respiration is a requirement to prevent labile C from building up in the plant? Overall, this statement is confusing and needs to be expanded on.

[revised manuscript text omitted]

---

## Editor Decision (ED1)

Dear Dr. Daniel Goll,

I will ask referees whether they are satisfied with your revision. But before that, you should address following defects in the revised manuscript.

(1) Figure 2 is duplicated, and correct Figure 6 is missing. Orders of figures 3 to 6 are off by one.

(2) In the caption of the Figure 1, ["mobile" (labile nitrogen)]: Switch the word mobile and labile.

(3) Figure 1: Denotations A, W, E, and N would be better to be unified to denotations in the main text ($C_A$, $C_W$, $C_E$, and $C_N$).

(4) Figure 1: Notifications "Non-lignified litter" and "Lignified litter" are only available for the nitrogen boxes. But, it would be better to put it on the carbon pools as well.

(5) Table 1 and main text. Variable identifier $F_{extr}$ is employed for both meaning of nitrogen input from excrements and nitrogen lost due to leaching. It's confusable.

(6) P6 L7~L8, $C_{lw}$ and $C_{la}$: It might be better to represent as $C_{l,w}$ and $C_{l,a}$.

(7) Table 3, $N_{lw}$ and $N_{la}$: It might be better to represent as $N_{l,w}$ and $N_{l,a}$.

(8) P8, Line 28, "decomposition rate of the corresponding YASSO carbon pool $C_H$, $k_H$": Here, $C_H$ should be in parenthesis.

(9) P13, L18: "appendix" should be replaced by "figure A1".

(10) P13, L26: "appendix" should be replaced by "table A1".

(11) P14, L1: "980-1030": Provide unit.

(12) P15 L1: A period missing.

(13) P15, L8 "25%": It should be 23% if table 6 provides correct values.

(14) P17, L4: "Figure 4d" would be replaced by "Figure 4e, f"

(15) Page 14 L11: A typo in "t hey".

(16) The caption of Figure A1: "(D)" would be replaced by "(B)".

---

## Author Response (AR3)

We thank the referee for the comments, and corrected all the points mentioned.

Best regards,
Daniel Goll

Page 2, Line 22, typo, "asses" (in tracked version )
    corrected
Figure 1, caption, " While the first two pools have no corresponding pool" is confusion. Do you mean no corresponding nitrogen pools? How about the labile pool?
    revised
It would be better to refer to appendix with their exact litter, e.g. appendix A
    corrected throughout the text
P22, L15, Is the citation in the right form with the bracket?
    brackets removed
Figure 3 seems to be not right, the result is the same as in Figure 2 instead of total C and primary production.
    this mistake is only in the tracked changes version, the main manuscript file is correct